# The chromatin regulator Ankrd11 controls cardiac neural crest cell-mediated outflow tract remodeling and heart function

Yana Kibalnyk[1,2], Elia Afanasiev[3], Ronan M. N. Noble[2,4], Adrianne E. S. Watson[1,2], Irina Poverennaya[5], Nicole L. Dittmann[1,6], Maria Alexiou[7], Kara Goodkey[1,2], Amanda A. Greenwell[2,8], John R. Ussher [2,8], Igor Adameyko [5,9], James Massey[10], Daniel Graf [1,2,7], Stephane L. Bourque [2,11], Jo Anne Stratton [3] & Anastassia Voronova [1,2,6,12] ✉

ANKRD11 (Ankyrin Repeat Domain 11) is a chromatin regulator and a causative gene for KBG syndrome, a rare developmental disorder characterized by multiple organ abnormalities, including cardiac defects. However, the role of ANKRD11 in heart development is unknown. The neural crest plays a leading role in embryonic heart development, and its dysfunction is implicated in congenital heart defects. We demonstrate that conditional knockout of *Ankrd11* in the murine embryonic neural crest results in persistent truncus arteriosus, ventricular dilation, and impaired ventricular contractility. We further show these defects occur due to aberrant cardiac neural crest cell organization leading to outflow tract septation failure. Lastly, knockout of *Ankrd11* in the neural crest leads to impaired expression of various transcription factors, chromatin remodelers and signaling pathways, including mTOR, BMP and TGF-β in the cardiac neural crest cells. In this work, we identify Ankrd11 as a regulator of neural crest-mediated heart development and function.

Congenital heart defects (CHDs) are the most common type of birth defect, with a global 1% incidence rate[1,2]. A significant portion of CHDs are conotruncal defects, which are caused by improper development of the heart outflow tract (OFT), an embryonic structure that connects the heart ventricles and pharyngeal arch arteries (PAAs)[3,4]. The OFT is remodeled from a single vessel into the base of the aorta and pulmonary artery during embryogenesis to separate the circulation of oxygenated and deoxygenated blood[3,4]. This process is orchestrated by interactions between different cell types of the embryonic heart: second heart field-derived cells, endocardial cells, and cardiac neural crest cells (CNCCs). Dysregulation of CNCC function is a primary cause of many

[1]Department of Medical Genetics, Faculty of Medicine & Dentistry, University of Alberta, Edmonton, AB T6G 2H7, Canada. [2]Women and Children's Health Research Institute, 5-083 Edmonton Clinic Health Academy, University of Alberta, 11405 87 Avenue NW, Edmonton, AB T6G 1C9, Canada. [3]Department of Neurology and Neurosurgery, Montreal Neurological Institute, McGill University, Montreal, QC H3A 2B4, Canada. [4]Department of Pediatrics, Faculty of Medicine & Dentistry, University of Alberta, Edmonton, AB T6G 2G3, Canada. [5]Department of Neuroimmunology, Center for Brain Research, Medical University of Vienna, 1090 Vienna, Austria. [6]Neuroscience and Mental Health Institute, Faculty of Medicine & Dentistry, University of Alberta, Edmonton, AB T6G 2E1, Canada. [7]Department of Dentistry, Faculty of Medicine & Dentistry, University of Alberta, Edmonton, AB T6G 2H7, Canada. [8]Faculty of Pharmacy & Pharmaceutical Sciences, University of Alberta, Edmonton, Alberta, Edmonton, AB T6G 2H1, Canada. [9]Department of Physiology and Pharmacology, Karolinska Institutet, 17177 Stockholm, Sweden. [10]Vizgen Inc., Cambridge, MA, USA. [11]Department of Anesthesiology & Pain Medicine, Faculty of Medicine & Dentistry, University of Alberta, Edmonton, AB T6G 2G3, Canada. [12]Department of Cell Biology, Faculty of Medicine & Dentistry, University of Alberta, Edmonton, AB T6G 2H7, Canada. ✉e-mail: voronova@ualberta.ca

conotruncal defects and contributes to many known multisystem developmental disorders[4,5].

CNCCs originate at the dorsal neural tube border and migrate into the pharyngeal arches and OFT, where they facilitate the remodeling of the vessels and form the smooth muscle cell lining of the arteries[3]. CNCCs populating the OFT cushions fuse at the OFT midline to create the aorticopulmonary (AP) septum, which progresses from the distal edge of the OFT until it fuses with the interventricular septum, creating two distinct vessels of the aorta and pulmonary artery. In mice, OFT septation occurs between embryonic day (E) 11.5 and E13.5[6].

Neural crest development requires intricate spatiotemporal regulation of gene expression. Most of the research has focused on signaling pathways and transcription factors, while epigenetic regulation has been relatively understudied[4,7-9]. Nevertheless, several epigenetic regulators, which are implicated in various congenital disorders, including Coffin-Siris (BRG1), CHARGE (CHD7), and Williams (BAZ1B) syndromes, have been found to be essential for proper CNCC development[7,10-12]. Chromatin regulators are critical for the complex spatiotemporal regulation of gene expression that orchestrates the OFT remodeling, although much of the mechanism behind the epigenetic regulation of neural crest-mediated heart development is still largely unclear[7].

ANKRD11 (Ankyrin repeat domain 11; previously known as ANCO1) is a chromatin regulator that interacts with histone acetylation modifying proteins, such as histone deacetylase HDAC3 and P/CAF (p300/CBP-associated factor) acetyltransferase complex subunits, to modulate gene expression[13-15]. Heterozygous pathogenic variants in *ANKRD11* or 16q24.3 microdeletions containing *ANKRD11* cause KBG syndrome (OMIM #148050), an autosomal dominant multisystem developmental disorder. Patients with KBG syndrome display global developmental delay, short stature, craniofacial defects, and intellectual disability[16-24]. About 40% of patients show cardiovascular defects, which include valve dysplasia and ventricular septal defects (VSD) as well as, less commonly, defects like aortic coarctation and patent ductus arteriosus[23,25,26]. Notably, these cardiovascular defects indicate a potential dysregulation of CNCC function[4,5]. However, the role of ANKRD11 in CNCCs or in cardiac development is not known.

Here, we demonstrate that ablation of *Ankrd11* in the murine neural crest leads to OFT septation failure, a severe congenital cardiac defect termed persistent truncus arteriosus (PTA), as well as ventricular dilation and impaired ventricular contractility. We further show that deletion of *Ankrd11* in the neural crest causes a delay in CNCC condensation in OFT cushions and failed AP septation as well as impairment of several signaling pathways, including mammalian target of rapamycin (mTOR), bone morphogenetic protein (BMP), and transforming growth factor-beta (TGF-β). This study identifies Ankrd11 as a crucial regulator of CNCC development and suggests CNCC dysregulation as a candidate cause for cardiac defects in KBG patients.

## Results
### Deletion of *Ankrd11* in the neural crest leads to OFT defects
To investigate if *Ankrd11* is expressed in CNCCs, we extracted *Ankrd11* mRNA expression data from a murine embryonic day (E) 9.5-E10.5 trunk neural crest single-cell RNA sequencing (scRNAseq) dataset[27]. The results showed that *Ankrd11* was broadly and robustly expressed in all neural crest cells, including early delaminating and non-cardiac *Sox10*+, *Ets1*+, *Foxd3*+ NCCs (Fig. 1a-c). Its expression was also observed in neural tube cells and central nervous system neurons, which supports and expands previous reports that demonstrate *Ankrd11* expression in the brain and craniofacial tissues[13,28] (Fig. 1b, c). *Ankrd11* expression persisted in neural crest cells in which *Foxd3* expression began to be downregulated, which signifies differentiation into cardiac derivatives (Fig. 1b). As all

CNCCs showed the presence of *Ankrd11* (Fig. 1c, d'), we further asked about the heterogeneity of its expression. Figure 1e shows that the majority (~64%) of CNCCs had a medium level of *Ankrd11* expression, while 22% of CNCCs had the lowest and 14% of CNCCs had the highest expression. However, the high expression cells were mostly located at the more differentiated side of the CNCC cluster (Fig. 1d-d'), indicating a positive correlation of *Ankrd11* expression along the pseudo time differentiation trajectory.

To determine how the loss of Ankrd11 in the neural crest affects cardiac development, we knocked out *Ankrd11* in the neural crest using the *Ankrd11*[fl/fl]; *Wnt1Cre*[2] mouse model (herein referred to as *Ankrd11*[ncko]), which was previously shown to have aberrant craniofacial development[28]. In these mice, Cre is expressed in pre-migratory neural crest cells under the *Wnt1* promoter, leading to the excision of *Ankrd11* exon 7 in neural crest cells and their derivatives[29,30]. Neural crest cell lineage tracing was performed by breeding in the *RosaYFP*[STOP] allele (*Ankrd11*[fl/fl]; *RosaYFP*[STOP/STOP]; *Wnt1Cre*[2]) leading to STOP cassette excision and expression of YFP (yellow fluorescent protein)[29,30]. At E11.5 the YFP distribution in both the conditional heterozygote (*Ankrd11*[nchet]: *Ankrd11*[fl/WT]; *Wnt1Cre*[2]; *RosaYFP*[STOP/STOP]) and conditional knockout (*Ankrd11*[ncko]: *Ankrd11*[fl/fl]; *Wnt1Cre*[2]; *RosaYFP*[STOP/STOP]) embryos were characteristic of the neural crest lineage (Fig. 1f, g), which includes the derivatives of the pharyngeal arches as well as the dorsal root ganglia of the peripheral nervous system[30].

To corroborate *Ankrd11* expression in control CNCCs and confirm the loss of *Ankrd11* in *Ankrd11*[ncko] CNCCs, we performed BaseScope, a single-molecule fluorescent in situ hybridization assay, with an *Ankrd11*-specific probe. In this assay, every detected fluorescent dot indicates a single *Ankrd11* mRNA molecule. Transverse sections of E11.5 OFT mesenchyme, which is predominantly composed of CNCCs[31], showed *Ankrd11* expression in ~80% of *Ankrd11*[nchet] (control) and ~20% of *Ankrd11*[ncko] cells ($p = 0.0055$; Fig. 1i-l', p). The residual *Ankrd11* signal in *Ankrd11*[ncko] samples may be due to the non-CNCC-derived cells of the OFT mesenchyme or incomplete Cre-mediated recombination. Furthermore, the non-neural crest-derived OFT myocardium and neural tube tissue showed comparable *Ankrd11* signals in control and *Ankrd11*[ncko] embryos (Fig. 1i"-l", n, o). Notably, the negative probe did not yield a signal under similar experimental and imaging conditions (Fig. 1m). *Ankrd11*-expressing cells in the *Ankrd11*[nchet] OFT mesenchyme were further stratified based on the number of *Ankrd11*+ dots per cell, which showed a range of *Ankrd11* expression (Fig. 1q). This is in line with the varied expression observed in the scRNAseq dataset (Fig. 1c-e). Together, these results support and extend our previous work, where we showed that *Ankrd11* is expressed in cranial neural crest cells[28].

Next, we examined how the loss of Ankrd11 in CNCCs affects heart morphology. Embryonic hearts at E18.5 were analyzed for anatomical anomalies using immunofluorescence analysis and micro-computed tomography (μCT). Homozygous ablation of *Ankrd11* in the neural crest was perinatal lethal, in agreement with ref. 28, so only embryonic timepoints were analyzed. In comparison to *Ankrd11*[WT] embryos, which expressed wild-type *Ankrd11* (*Ankrd11*[WT/WT]; *Wnt1Cre*[2]; *RosaYFP*[STOP/STOP], $n = 9$) and *Ankrd11*[nchet] embryos ($n = 29$), all *Ankrd11*[ncko] embryos ($n = 40$) exhibited a persistent truncus arteriosus (PTA), where the OFT failed to septate into the pulmonary trunk and aorta and remained a single vessel (Fig. 2a-c). This was accompanied by a ventricular septal defect (VSD). Since neural crest-specific ablation of one copy of *Ankrd11* (*Ankrd11*[nchet]) did not cause apparent OFT septation defects, further analysis was done by comparing *Ankrd11*[ncko] to both *Ankrd11*[nchet] and/or *Ankrd11*[WT] embryos. Notably, the thymus, an organ located above the heart that receives neural crest contribution[32], did not differ in size between *Ankrd11*[nchet] and *Ankrd11*[ncko] embryos (Fig. S1a-c). This demonstrates the specificity of neural crest-mediated ablation of *Ankrd11* on distinct organ morphogenesis, such as the heart.

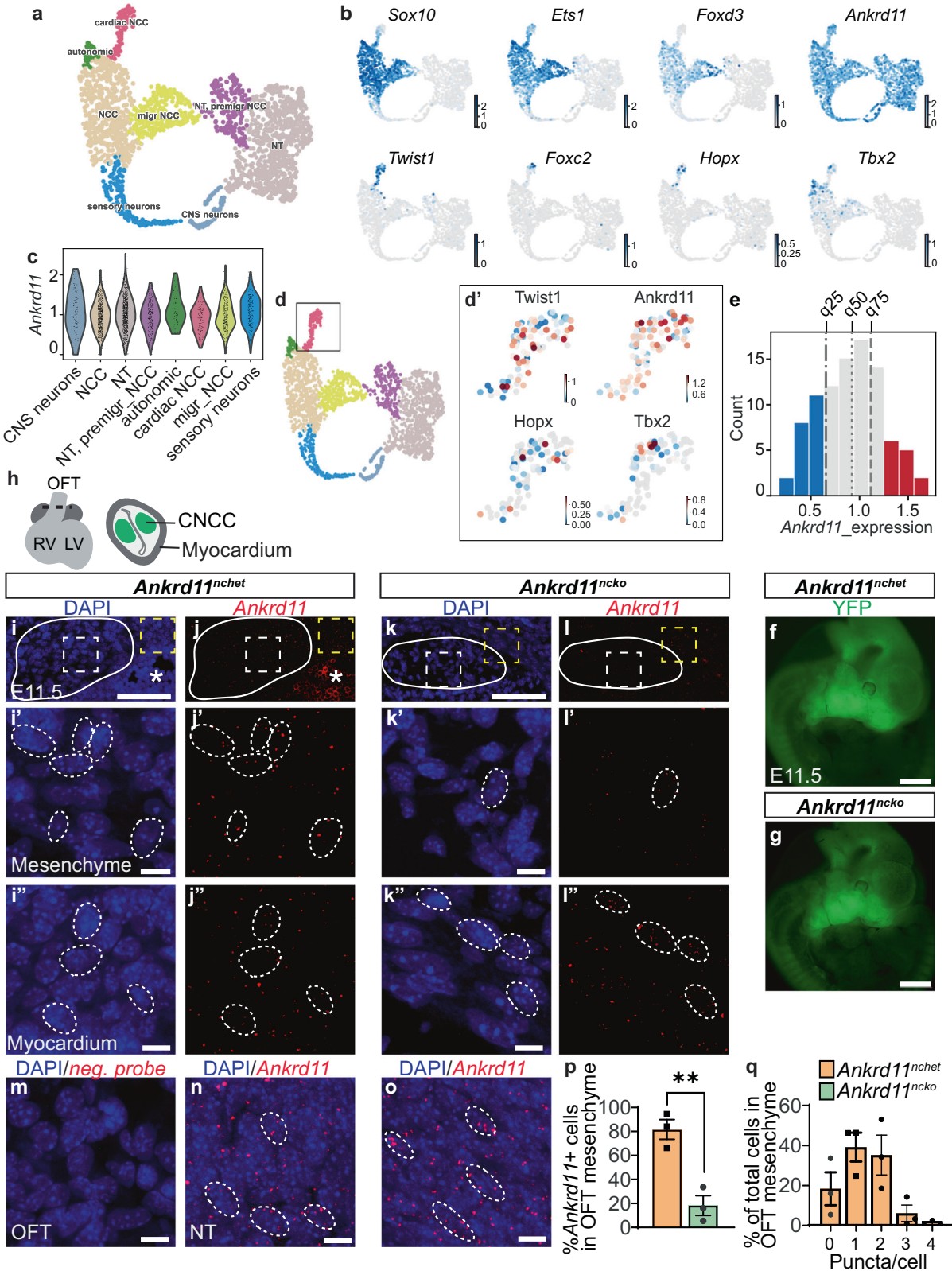

Additional analysis revealed that *Ankrd11^{ncko}* embryos had dysplastic truncal valves, visualized by PDGFRα immunostaining[33] (Fig. S1d–i), with 3 out of 5 analyzed hearts showing enlarged valves when compared to *Ankrd11^{nchet}* (Fig. S1f, g). Notably, analysis of the αSMA + (alpha-smooth muscle actin) smooth muscle layer of the

truncal artery did not show a deficiency of CNCC-derived smooth muscle cells compared to the aortic and pulmonary arteries (Fig. S1j–p). Although the *Ankrd11^{ncko}* distal region showed a decreased proportion of smooth muscle cells that are CNCC-derived (% αSMA + YFP +/αSMA +; *p* = 0.010763; Fig. S1n), this was not due to their

**Fig. 1 | *Ankrd11* is expressed in cardiac neural crest cells and ablated in the *Ankrd11*^ncko^ mouse model. a** UMAP embedding of mouse trunk neural crest and neural tube lineage at E9.5 and E10.5 (total 2121 cells). Data were re-analyzed from ref. 27. **b** mRNA expression of *Ankrd11* and key markers for migratory and cardiac neural crest, abundance indicated by color intensity. **c** *Ankrd11* expression level distribution in different cell types of mouse trunk neural crest and neural tube lineage at E9.5 and E10.5. **d–d'** Expression of *Ankrd11* and key markers in cardiac NCC cluster (92 cells). **e** *Ankrd11* expression level distribution in cardiac NCC cluster. Dash-dotted, dotted, and dashed lines refer to the first (blue), second (gray), and third (red) quartiles (q), respectively. **f, g** Representative images of *Ankrd11*^nchet^ (**f**) and *Ankrd11*^ncko^ (**g**) embryos at E11.5 showing YFP fluorescence. **h** Schematic of OFT transverse section at E11.5. **i–o** Representative images of *Ankrd11*^nchet^ (*Ankrd11*^fl/WT^;*Wnt1Cre*^2^, **i–j"**, **m**, **n**, and *Ankrd11*^ncko^ (*Ankrd11*^fl/fl^;*Wnt1Cre*^2^, **k–l"**, **o** E11.5 OFT (**i–m**) or neural tube (**n, o**) sections processed for Basescope with *Ankrd11* exon 7 mRNA probe (**i–l"**, **n, o**) and control negative probe (**m**) (red), and

counterstained for cell nuclei (DAPI; blue). Solid outlines: OFT mesenchyme. * autofluorescent red blood cells. **i'–l'** High magnification images of OFT mesenchyme (white dashed boxes in **i–l**). Dashed outlines: *Ankrd11*+ cells. **i"–l"** High magnification images of OFT myocardium (yellow dashed boxes in **i–l**). Dashed outlines: *Ankrd11*+ cells. **n, o** High magnification images of neural tube (NT) regions. Dashed outlines: *Ankrd11*+ cells. **p** Quantification for i-l for % *Ankrd11*+ cells in *Ankrd11*^nchet^ (orange) and *Ankrd11*^ncko^ (green) OFT mesenchyme. Two-tailed unpaired *t*-test (*p* = 0.0055). **q** Quantification of p for the distribution of *Ankrd11*+ puncta/cell in *Ankrd11*^nchet^ OFT mesenchyme. **p < 0.01. Graphs represent mean ± s.e.m, *n* = 3 *Ankrd11*^nchet^ and *n* = 3 *Ankrd11*^ncko^ samples from two independent litters. Scale bars: 1 mm (**f, g**), 100 μm (**i–l**), 10 μm (**i'–o**). CNCC cardiac neural crest cells, CNS central nervous system, LV left ventricle, migr migratory, NCC neural crest cells, NT neural tube, OFT outflow tract, premigr pre-migratory, RV right ventricle. Source data are provided as a Source Data file.

---

decreased number (#YFP + αSMA+; Fig. S1o), but rather due to an increase in non-CNCC-derived smooth muscle cells (#YFP-αSMA +; *p* = 0.034171; Fig. S1p).

μCT imaging and subsequent 3D reconstruction confirmed PTA in *Ankrd11*^ncko^ hearts, with the truncal artery originating from the right ventricle in all cases (Fig. 2d, e). *Ankrd11*^ncko^ ventricles were ~1.8-fold larger (*p* = 0.0057) than *Ankrd11*^ctrl^ (*Ankrd11*^fl/fl^, *Ankrd11*^fl/WT^, and *Ankrd11*^nchet^) (Fig. 2d–f), although ventricular wall thickness was unchanged (Fig. 2g). More detailed analysis of vessel branching identified an interrupted aortic arch phenotype to accompany the PTA (Fig. 2d', e')[34]. These results show that *Ankrd11*^ncko^ embryos had severe OFT and heart development defects.

### *Ankrd11*^ncko^ hearts show impaired contractility and function

To investigate whether the observed abnormalities affected cardiac function, we performed in utero echocardiography at E18.5[35] (Fig. 2 & S1; Supplementary Movie 1 & 2). In agreement with wall thickness measurements through μCT analysis, echocardiography confirmed that there was no difference in wall thicknesses between *Ankrd11*^ncko^ and control hearts (Fig. S1r). Fetal heart rate was also unchanged (Fig. S1q). Chamber size was increased in the *Ankrd11*^ncko^ hearts compared to controls (Fig. 2h–m). Specifically, the left ventricular (LV) and right ventricular (RV) areas in systole were increased by ~3-fold (*p* = 0.012655) and ~2.6-fold (*p* = 0.013974), respectively (Fig. 2h–l). Furthermore, LV and RV diameter in systole and RV diameter in diastole was increased by ~1.9-fold (*p* = 0.018753), ~2.4-fold (*p* = 0.005594), and ~1.7-fold (*p* = 0.015199), respectively (Fig. S1s, t). LV and RV area in diastole showed a non-significant trending increase in *Ankrd11*^ncko^ embryos compared to controls (Fig. 2m; *p* = 0.06) and LV diameter in diastole showed a non-significant trending increase in *Ankrd11*^ncko^ embryos compared to controls (Fig. S1t; *p* = 0.18). Finally, measures of cardiac contractility were reduced in *Ankrd11*^ncko^ hearts when compared to controls. LV and RV percent area change decreased by ~1.4-fold (*p* = 0.000121) and ~1.3-fold (*p* = 0.000974), respectively (Fig. 2n), and LV and RV fractional shortening decreased by ~1.7-fold (*p* = 0.0406) and ~1.9-fold (*p* = 0.0307), respectively (Fig. S1u, v). These results indicate that *Ankrd11* ablation in the neural crest leads to cardiac functional defects, specifically impaired ventricular contractility.

### *Ankrd11*^ncko^ CNCCs show a mild defect in migration into the OFT

To determine how the loss of Ankrd11 affects CNCC fates, we first analyzed CNCC migration, proliferation, and apoptosis in the OFT at E10.5, prior to OFT remodeling[6] (Fig. S2). Analysis of CNCC distribution along the proximal-distal gradient in the E10.5 OFT revealed ~1.4-fold fewer (*p* = 0.038597) YFP+ CNCCs specifically in the medial region of *Ankrd11*^ncko^ OFTs compared to control (*Ankrd11*^nchet^) (Fig. S2a–g). No changes were found in the proliferative index of CNCCs expressing Ki67, a proliferation

marker[28] (%Ki67+YFP + /YFP+; Fig. S2a'–f', h), or in the proportion of apoptotic YFP + TUNEL+ (terminal deoxynucleotidyl transferase dUTP nick end labeling) CNCCs (Fig. S2a"–f", i). This suggests that the deficit in CNCC numbers in *Ankrd11*^ncko^ OFTs was most likely due to a modestly reduced CNCC migration into the OFT rather than abnormal proliferation or apoptosis.

### *Ankrd11*^ncko^ OFTs show failed septation

To determine the role of Ankrd11 in OFT remodeling, we analyzed the OFT at E11.5 and E12.5, when OFT septation begins[6]. At E11.5, OFTs in control hearts began to show significant remodeling, which was absent in the OFTs of *Ankrd11*^ncko^ hearts (Fig. 3). In the distal region of E11.5 control OFTs, YFP+ CNCCs have formed the AP septum, which was densely filled by parallel F-actin filaments, visualized using phalloidin staining (Fig. 3a–c). In contrast, the AP septum was absent in the anatomically matched regions of E11.5 *Ankrd11*^ncko^ OFTs (Fig. 3d, e). However, by E12.5, *Ankrd11*^ncko^ OFTs formed an AP septum only at the very distal edge of the OFT, with the rest of the OFT unseptated (Fig. 3j–t).

In medial regions of E11.5 control OFTs, CNCCs were organized into two tightly condensed columns within the cardiac cushions, visualized by spiral arrangements of CNCC nuclei and F-actin filaments in relation to the OFT center (Fig. 3f–g'). This condensation precedes CNCC migration to the OFT midline and their fusion to create the AP septum[36]. In contrast, in the anatomically matched regions of the *Ankrd11*^ncko^ OFT, CNCCs showed less organization and condensation, with more varied orientation of nuclei and actin filaments in relation to the OFT cushion center compared to the control (Fig. 3h–i'). However, by E12.5 *Ankrd11*^ncko^ medial region CNCCs showed more organized spiral nuclear and actin filament orientation that was more similar to control (Fig. 3r–u'). This was not observed in the unseptated distal region of E12.5 *Ankrd11*^ncko^ OFTs, where the CNCC arrangement remained disorganized (Fig. 3n–q). Since the *Ankrd11*^ncko^ OFTs remained unseptated by E18.5 (Fig. 2), this suggests that the septation process stalled in this region.

CNCC orientation to cushion center was quantified by binning the cells into categories by their nuclear orientation relative to the OFT cushion center (90°-70°, 69°-50°, 49°-30°, 29°-0°; Fig. 3v), adapted from ref. 36. Nuclei oriented perpendicularly to the cushion center are closer to 90° while nuclei oriented in parallel are closer to 0° (Fig. 3v). Since the remodeled distal region of control OFTs was morphologically distinct from the unremodeled *Ankrd11*^ncko^ OFTs (Fig. 3b–e), we chose to compare medial regions for all subsequent quantifications, which had more comparable morphology (Fig. 3f–i). In control OFTs at E11.5 and E12.5, the 90°-70° bin contained the greatest number of YFP+ CNCCs (Fig. 3w–x). In contrast, all bins in the E11.5 *Ankrd11*^ncko^ analysis had approximately similar numbers of YFP+ CNCCs (Fig. 3w). Compared to the control, E11.5 *Ankrd11*^ncko^ OFTs had a notably decreased proportion of CNCCs in the 90°–70°

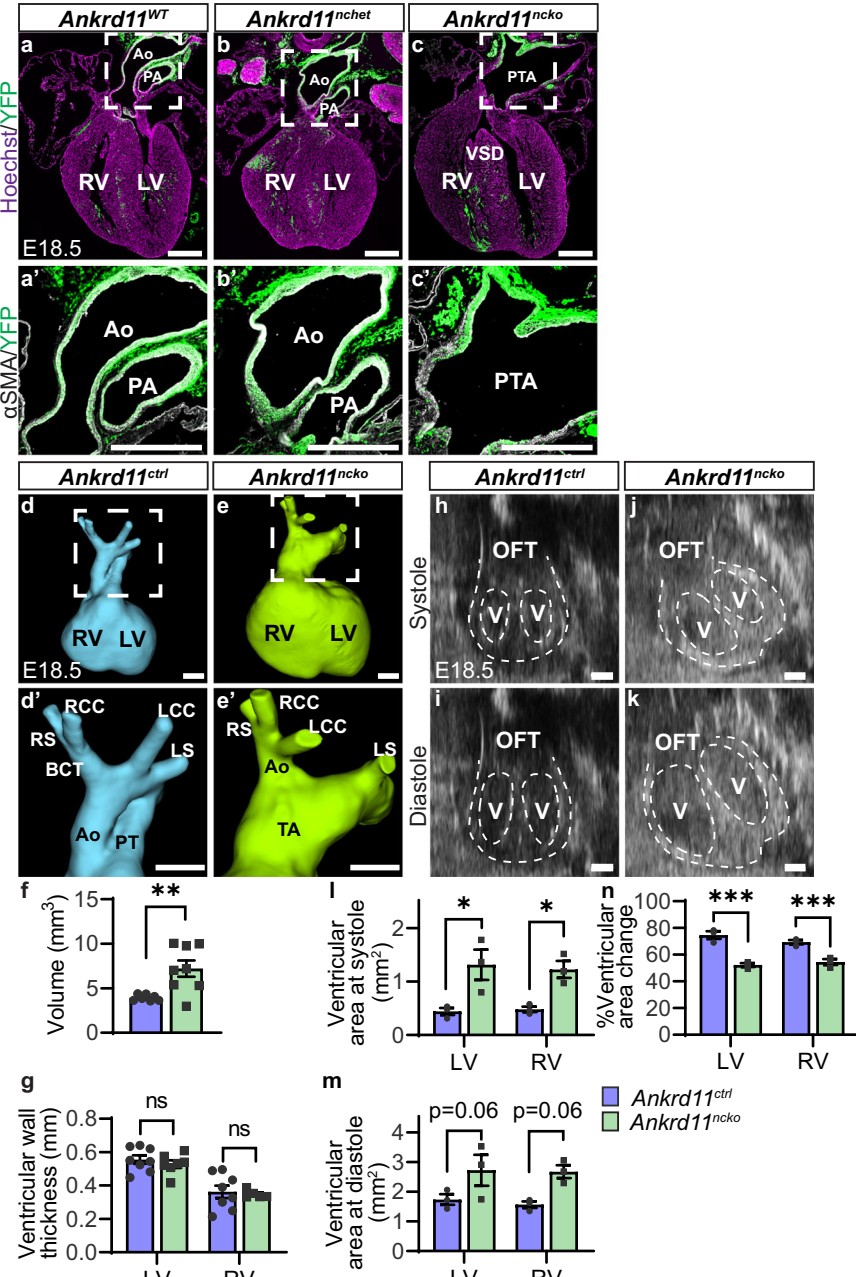

**Fig. 2 | Conditional loss of *Ankrd11* in the neural crest leads to outflow tract defects and decreased ventricular function at E18.5.** *Related to* Supplemental Fig. 1 *and the associated movie files.* **a**–**c** Representative images of *Ankrd11^WT^* (*Ankrd11^WT/WT^;Wnt1Cre²*, **a**), *Ankrd11^nchet^* (*Ankrd11^fl/WT^;Wnt1Cre²*, **b**) and *Ankrd11^ncko^* (*Ankrd11^fl/fl^;Wnt1Cre²*, **c**) E18.5 heart coronal sections, immunostained for CNCC derivatives (YFP; green) and counterstained for cell nuclei (Hoechst; magenta). **a'**–**c'.** High magnification images of OFT regions (white dashed boxes in **a**–**c**) from the same or adjacent sections immunostained for alpha-smooth muscle actin (αSMA; white) and CNCC derivatives (YFP; green). **d**–**e'.** Representative 3D reconstructions of μCT scans of E18.5 *Ankrd11^ctrl^* (*Ankrd11^fl/fl^, Ankrd11^fl/WT^* and *Ankrd11^nchet^* **d**) and *Ankrd11^ncko^* (**e**) hearts and magnified images (**d'**, **e'**) of the OFT regions (from white dashed boxes in **d**, **e**). **f**, **g** Quantification of d-e for total ventricular volumes (**f**) and ventricular wall thicknesses (**g**) from *Ankrd11^ctrl^* (blue) and *Ankrd11^ncko^* (green) embryos. Two-tailed unpaired *t*-test (**f**, *p* = 0.0057); two-tailed multiple *t*-tests with Holm-Sidak multiple comparisons test (**g**). **h**–**k** Representative B-mode long-axis images of in utero echocardiography of *Ankrd11^ctrl^* (*Ankrd11^fl/fl^, Ankrd11^fl/WT^*, and

*Ankrd11^nchet^*, **h**, **i**) and *Ankrd11^ncko^* (**j**, **k**) embryos during systole (**h**, **j**) and diastole (**i**, **k**). Dashed lines outline the heart and ventricles in each image. **l**–**n** Quantification of the ventricular area during systole (**l**), diastole (**m**), and percent area change between systole and diastole (**n**) in the left (LV) and right (RV) ventricles from *Ankrd11^ctrl^* and *Ankrd11^ncko^* embryos. Two-tailed multiple *t*-tests with Holm-Sidak multiple comparisons test (**l**, LV *p* = 0.012655, RV *p* = 0.013974; **n**, LV *p* = 0.000121, RV *p* = 0.000974). ns: not significant, \**p* < 0.05, \*\* *p* < 0.01, \*\*\**p* < 0.001. Graphs represent mean ± s.e.m; *n* = 7 *Ankrd11^ctrl^* and 8 *Ankrd11^ncko^* (**f**); *n* = 8 *Ankrd11^ctrl^*, 7 *Ankrd11^ncko^* LV, 5 *Ankrd11^ncko^* RV (**g**); *n* = 3 *Ankrd11^ctrl^* and *n* = 3 *Ankrd11^ncko^* (**l**–**n**) samples. Embryos were taken from 3 independent litters. Scale bars: 500 μm. Ao aorta, BCT brachiocephalic trunk, LCC left common carotid artery, LS left subclavian artery, LV left ventricle, OFT outflow tract, PA pulmonary artery, RCC right common carotid artery, RS right subclavian artery, RV right ventricle, TA truncal artery, V ventricle, VSD ventricular septal defect. Source data are provided as a Source Data file.

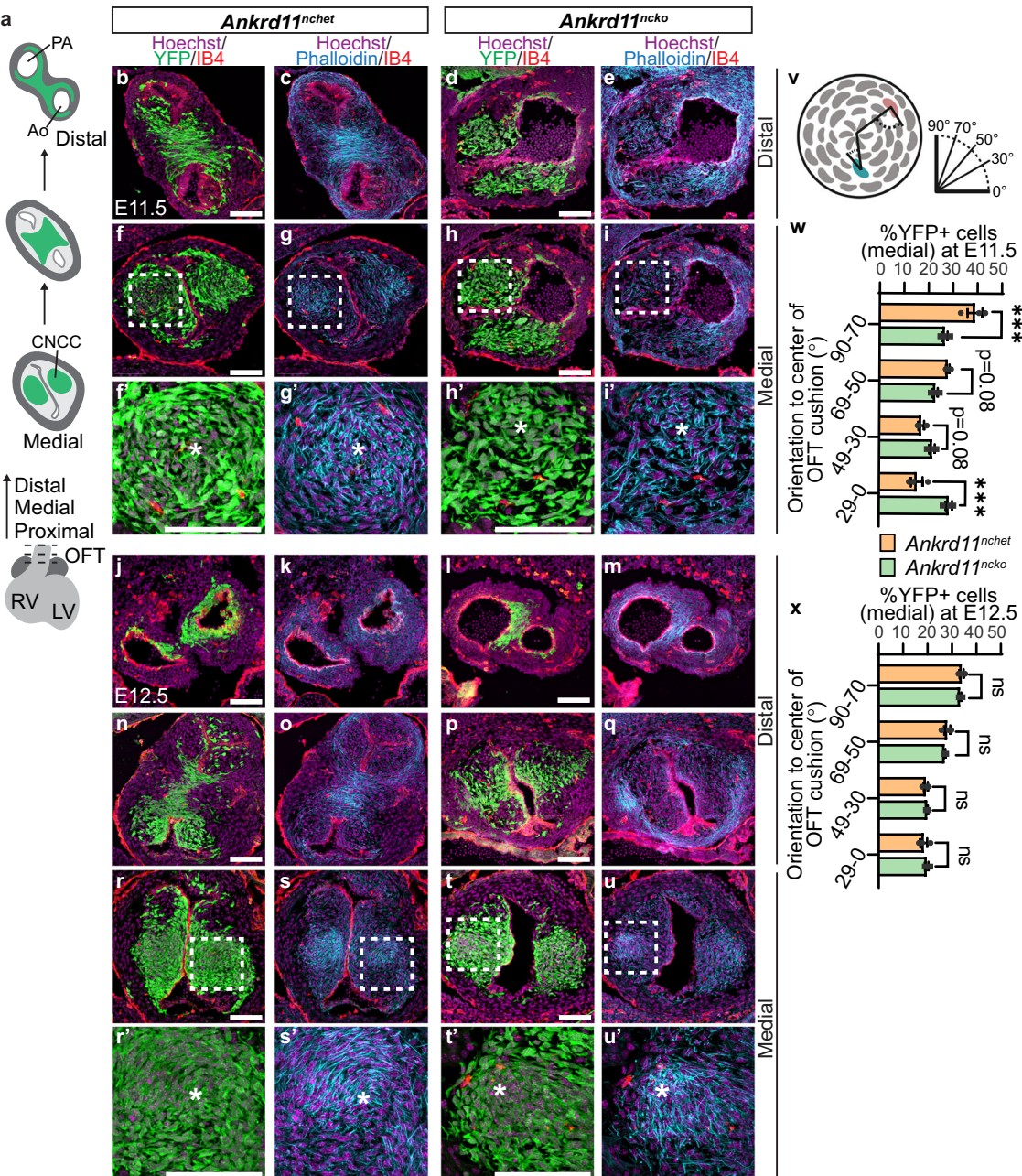

**Fig. 3 | Conditional loss of *Ankrd11* in the neural crest causes delayed CNCC organization and failed OFT septation at E11.5-E12.5. *Related to* Supplemental Fig. 2. a** Schematic of OFT transverse sections undergoing different stages of remodeling along the proximal-distal axis from E11.5 to E12.5. In the proximal and medial OFT, CNCCs in green are condensed into tightly organized columns. In the distal region, streams of CNCCs from both OFT cushions migrated medially and fused to form the aorticopulmonary (AP) septum and two distinct vessels of the aorta (Ao) and pulmonary artery (PA). **b–e** Representative images of E11.5 *Ankrd11^nchet^* (**b, c**) and *Ankrd11^ncko^* (**d, e**) distal OFT transverse sections immunostained for CNCC and their derivatives (YFP; green) and F-actin (phalloidin; blue) and counterstained for nuclei (Hoechst; magenta). **f–i'**. Representative images of E11.5 *Ankrd11^nchet^* (**f, g**) and *Ankrd11^ncko^* (**h, i**) medial OFT sections. White dashed boxes are shown at higher magnification in **f'–i'**. * OFT cushion center. **j–q** Representative images of E12.5 *Ankrd11^nchet^* (**j, k, n, o**) and *Ankrd11^ncko^* (**l, m, p, q**) distal OFT sections.

**r–u'** Representative images of E12.5 *Ankrd11^nchet^* (**r, s**) and *Ankrd11^ncko^* (**t, u**) medial OFT sections. White dashed boxes are shown at higher magnification in **r'–u'**. **v** Schematic of *Ankrd11^nchet^* CNCC nuclei orientation within an OFT cushion at E11.5-E12.5. Nuclei oriented perpendicularly to the cushion center (pink cell) are closer to 90° while nuclei oriented in parallel (teal cell) are closer to 0°. Method adapted from ref. 36. Created with BioRender.com released under a Creative Commons Attribution-NonCommercial-NoDerivs 4.0 International license. **w, x** Quantification of CNCC nuclear orientation to OFT cushion center at E11.5 and E12.5 in medial regions between *Ankrd11^nchet^* (orange) and *Ankrd11^ncko^* (green) genotypes. Two-tailed multiple t-tests with Holm-Sidak multiple comparisons test (w, 90-70 $p = 0.000226$, 29-0 $p = 0.000148$). ns: not significant; $p > 0.05$ ***$p < 0.001$. Graphs represent mean ± s.e.m; $n = 3$ *Ankrd11^nchet^* and $n = 3$ *Ankrd11^ncko^* samples from at least 2 independent litters. Scale bars: 100 μm. Source data are provided as a Source Data file.

bin ($p = 0.000226$) and an increased proportion of CNCCs in the 29°–0° bin ($p = 0.000148$; Fig. 3w). This indicates that *Ankrd11^ncko^* CNCCs showed impaired cell organization at E11.5. However, by E12.5, the nuclear orientation of *Ankrd11^ncko^* CNCCs was comparable

to the control (Fig. 3x), suggesting that *Ankrd11^ncko^* CNCCs showed a delay in condensation. These results show that *Ankrd11* ablation in the neural crest causes delayed CNCC organization in the OFT and failed AP septum formation.

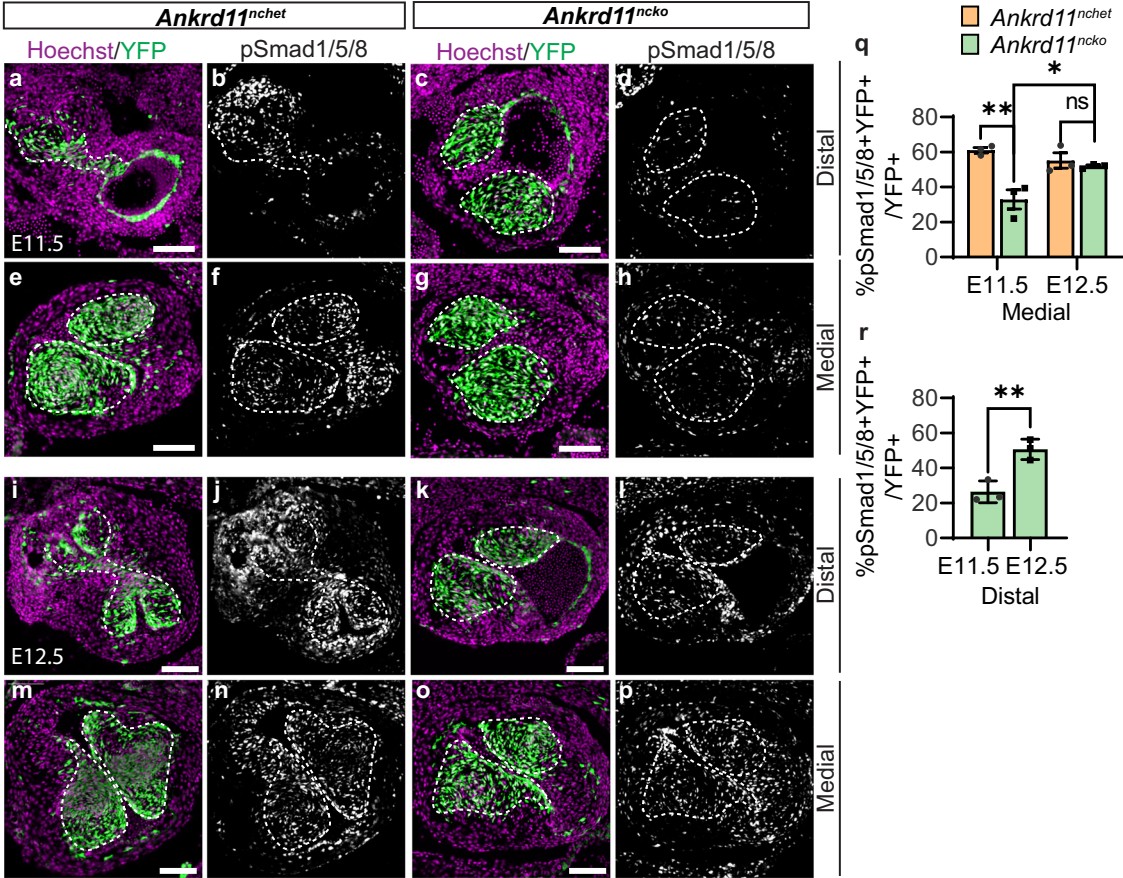

**Fig. 4 | Conditional loss of *Ankrd11* in the neural crest causes delayed BMP signaling in OFT cushions. a–p** Representative images of *Ankrd11^{nchet}* (**a**, **b**, **e**, **f**, **i**, **j**, **m**, **n**) and *Ankrd11^{ncko}* (**c**, **d**, **g**, **h**, **k**, **l**, **o**, **p**) distal and medial OFT cushions (outlined with white dashed lines) at E11.5 (**a–h**) and E12.5 (**i–p**) immunostained for CNCCs (YFP; green), pSmad1/5/8 (white) and counterstained for nuclei (Hoechst, magenta). **q**, **r** Quantification of **a–p** for percent of pSmad1/5/8+ CNCCs (% pSmad1/5/8 + YFP + /YFP+ cells) in *Ankrd11^{nchet}* (orange) and *Ankrd11^{ncko}* (green) genotypes in the medial region, and in the *Ankrd11^{ncko}* genotype the distal region, at E11.5 and E12.5. Two-way ANOVA with Sidak's multiple comparisons test (**q**, E11.5 *Ankrd11^{nchet}* vs. E11.5 *Ankrd11^{ncko}* $p = 0.0035$, E11.5 *Ankrd11^{ncko}* vs. E12.5 *Ankrd11^{ncko}* $p = 0.035$); two-tailed unpaired *t*-test (**r**, $p = 0.0078$). Graphs represent mean ± s.e.m; $n = 3$ *Ankrd11^{nchet}* and $n = 3$ *Ankrd11^{ncko}* samples from at least 2 independent litters. ns: not significant, *$p < 0.05$, **$p < 0.01$. Scale bars: 100 μm. Source data are provided as a Source Data file.

## *Ankrd11^{ncko}* CNCCs have impaired signaling pathways

To understand the mechanism behind the delay in the organization of Ankrd11-deficient CNCC and consequent AP septation failure, we sought to explore the signaling pathways that are known to regulate OFT development. BMP signaling has been shown to be activated along a distal-proximal gradient and impairment of its modulation dysregulated the timing of CNCC condensation[36]. To test for BMP signaling changes, we used immunostaining for phosphorylated Smad1/5/8, the intracellular effectors of Smad-dependent BMP signaling[36–38]. We observed pSmad1/5/8 signal in the medial and distal regions in control and *Ankrd11^{ncko}* OFT cushions (Fig. 4a–p). In control medial OFT cushions at E11.5, we detected ~61% of YFP+ CNCCs with high pSmad1/5/8 signal (% pSmad1/5/8 + YFP + /YFP+ cells) (Fig. 4e–f, q). In contrast, we observed only ~33% of *Ankrd11^{ncko}* pSmad1/5/8+ CNCCs in the medial region, a ~1.9-fold ($p = 0.0035$) decrease compared to control (Fig. 4g–h, q). However, by E12.5, we observed ~52% of *Ankrd11^{ncko}* pSmad1/5/8+ CNCCs in the medial region, a ~1.6-fold ($p = 0.035$) increase compared to *Ankrd11^{ncko}* medial region at E11.5 and statistically comparable ($p = 0.9908$) to control medial region at E12.5 (Fig. 4m–q). For the distal region, we only analyzed *Ankrd11^{ncko}* cardiac cushions as controls have formed a septum in this area. At E11.5, we detected ~26% of *Ankrd11^{ncko}* pSmad1/5/8+ CNCCs (Fig. 4c, d, r) compared to ~51% of *Ankrd11^{ncko}* CNCCs at E12.5 in the unseptated distal region (Fig. 4k, l, r), which represents a ~1.9-fold ($p = 0.0078$) increase compared to the *Ankrd11^{ncko}* distal region at E11.5. This suggests that

*Ankrd11^{ncko}* CNCCs show delayed BMP signaling, with reduced pSmad1/5/8 levels at E11.5 and normal levels at E12.5.

The mTORC1 signaling pathway has been recently implicated in OFT development, including the regulation of actin dynamics required for cell migration[39]. We analyzed phosphorylated ribosomal protein S6 (pS6) of the 40 s ribosomal subunit, a well-accepted readout of mTORC1 activation, and compared it to total S6 in E11.5 OFTs[40,41]. We observed pS6 and total S6 signals in the medial and distal regions in control and *Ankrd11^{ncko}* OFT cushions (Fig. 5a–p, S3a–p). In control medial OFT cushions at E11.5, we observed ~63% pS6+ CNCCs (% pS6 + S6 + YFP + /S6 + YFP+ cells) (Fig. 5e, f, q). In contrast, we observed only ~37% pS6+ *Ankrd11^{ncko}* CNCCs in the medial region, a ~1.7-fold ($p = 0.0108$) decrease compared to the control (Fig. 5e–h, q). However, by E12.5, we observed ~60% pS6+ *Ankrd11^{ncko}* CNCCs in the medial region, a ~1.6-fold ($p = 0.0182$) increase compared to *Ankrd11^{ncko}* medial region at E11.5, and statistically comparable ($p = 0.2069$) to control medial region at E12.5 (Fig. 5m–q). As described above, for the distal region, we only analyzed *Ankrd11^{ncko}* cardiac cushions as controls have formed a septum in this area. In the distal region, we detected ~48% pS6+ *Ankrd11^{ncko}* CNCCs at E11.5 (Fig. 5c, d, r), compared to ~76% pS6+ *Ankrd11^{ncko}* CNCCs at E12.5 in the unseptated distal region (Fig. 5k, l, r), a ~1.6-fold ($p = 0.0033$) increase compared to E11.5. This suggests that *Ankrd11^{ncko}* CNCCs show delayed mTORC1 signaling, with reduced pS6 levels at E11.5 and normal pS6 levels at E12.5.

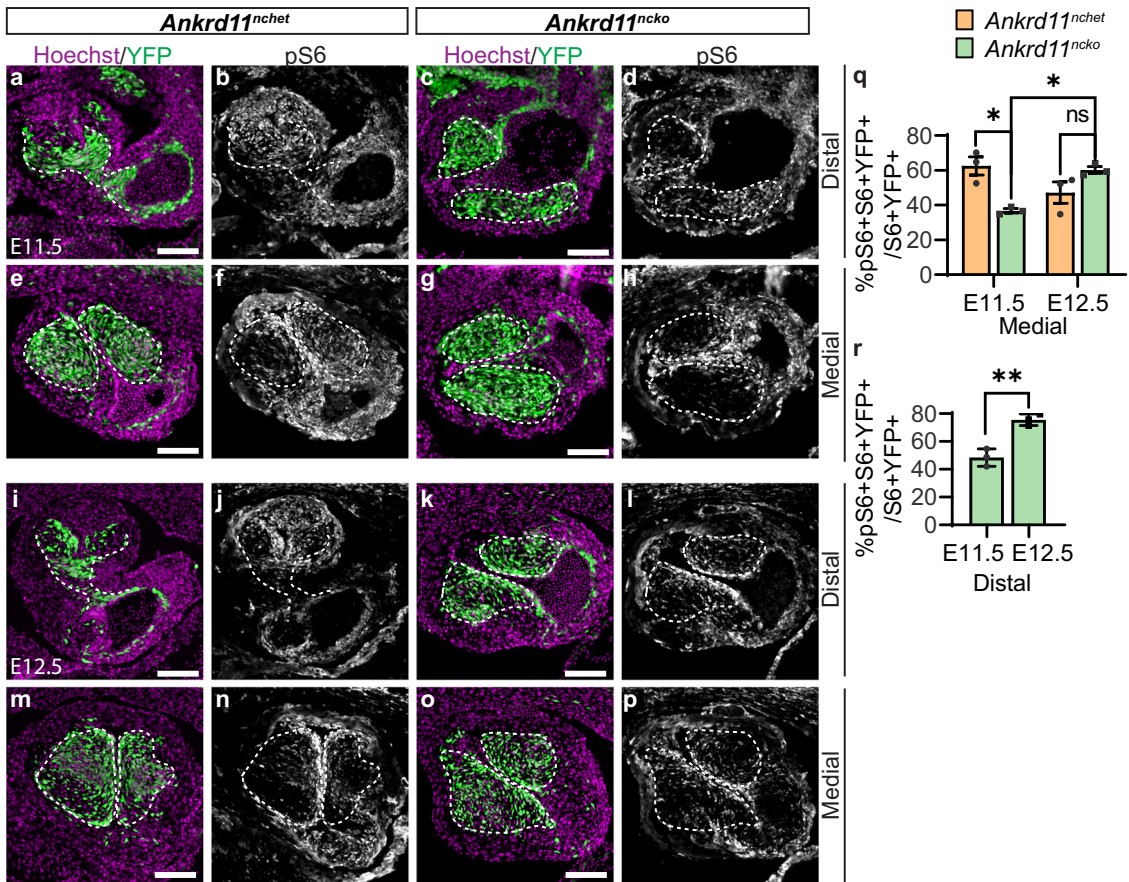

**Fig. 5 | Conditional loss of *Ankrd11* in the neural crest causes delayed mTOR signaling in OFT cushions.** *Related to* Supplemental Fig. 3. **a–p** Representative images of *Ankrd11^nchet^* (**a**, **b**, **e**, **f**, **i**, **j**, **m**, **n**) and *Ankrd11^ncko^* (**c**, **d**, **g**, **h**, **k**, **l**, **o**, **p**) distal and medial OFT cushions (outlined with white dashed lines) at E11.5 (**a–h**) and E12.5 (**i–p**) immunostained for CNCCs (YFP; green), pS6 (white) and counterstained for nuclei (Hoechst, magenta). **q, r** Quantification of **a–p** for percent of pS6 CNCCs (% pS6 + S6 + YFP + /S6 + YFP+ cells) in *Ankrd11^nchet^* (orange) and *Ankrd11^ncko^* (green) medial OFT (**q**), and in the *Ankrd11^ncko^* distal OFT (**r**), at E11.5 and E12.5. Two-way ANOVA with Sidak's multiple comparisons test (**q**, E11.5 *Ankrd11^nchet^* vs. E11.5 *Ankrd11^ncko^* p = 0.0108, E11.5 *Ankrd11^ncko^* vs. E12.5 *Ankrd11^ncko^* p = 0.0182); two-tailed unpaired *t*-test (**r**, p = 0.0033). Graphs represent mean ± s.e.m; n = 3 *Ankrd11^nchet^* and n = 3 *Ankrd11^ncko^* samples from at least 2 independent litters. ns: not significant, *p < 0.05, **p < 0.01. Scale bars: 100 μm. Source data are provided as a Source Data file.

The Smad2/3-dependent TGF-β pathway is also known to be involved in OFT development, with ablation of several pathway components causing OFT septation defects[42–45]. Therefore, to test for abnormalities in TGF-β signaling, we used immunostaining for phosphorylated Smad2/3, the intracellular effectors of Smad-dependent TGF-β signaling[45]. We observed pSmad2/3 signal throughout control and *Ankrd11^ncko^* OFT cushions (Fig. 6a–p). In control medial OFT cushions at E11.5, we observed that the parietal cushion (pc), positioned in the top-right of the OFT, had significantly higher pSmad2/3 signaling intensity compared to the septal cushion (sc), positioned in the bottom-left OFT, comprising ~55% CNCCs with high pSmad2/3 signal (% pSmad2/3 + YFP + /YFP+ cells) and ~33% pSmad2/3+ CNCCs, respectively (p = 0.030585; Fig. 6e, f, q). This asymmetrical pSmad2/3 staining persisted in E12.5 control medial OFT but was less pronounced compared to E11.5 (Fig. 6m, n, s). In contrast, we observed that parietal and septal cushions in the E11.5 *Ankrd11^ncko^* medial OFT had similar levels of pSmad2/3 signaling, with ~51% and ~48% pSmad2/3+ CNCCs in the parietal and septal cushions, respectively (Fig. 6g, h, q). However, by E12.5, *Ankrd11^ncko^* medial OFT showed a higher proportion (~63%) of pSmad2/3+ CNCCs in the parietal cushion compared to ~42% in the septal cushion (p = 0.002271; Fig. 6o, p, s). As described above, for the distal region, we only analyzed *Ankrd11^ncko^* cardiac cushions as controls have formed a septum in this area. In the distal region, at both E11.5 and E12.5, we detected comparable levels of pSmad2/3+ CNCCs between *Ankrd11^ncko^* parietal and septal cushions (Fig. 6c, d, k, l, r, t). These results suggest that *Ankrd11^ncko^* OFTs show delayed establishment of TGF-β signaling asymmetry, with abnormal symmetric pSmad2/3 pattern in parietal and septal cushions at E11.5, and normal asymmetric pattern at E12.5.

We next sought to find other factors that may be asymmetrically expressed in OFT cushions. We focused on Cellular retinoic acid binding protein 2 (Crabp2), a protein that transports retinoic acid from the cytoplasm to the nucleus, which facilitates retinoic acid signaling[46]. In control medial OFT cushions at E11.5, we observed that the parietal cushion (pc) had lower Crabp2 levels compared to the septal cushion (sc), comprising 9% CNCCs with high Crabp2 signal (% Crabp2+YFP + /YFP+ cells) and ~19% Crabp2+ CNCCs, respectively (p = 0.046044; Fig. S4e, f, i). In contrast, *Ankrd11^ncko^* OFTs displayed asymmetric Crabp2 levels only in the distal region with ~22% Crabp2+ CNCCs in the parietal cushion compared to ~39% Crabp2+ CNCCs in the septal cushion (p = 0.0126; Fig. S4c, d, j). This suggests that *Ankrd11^ncko^* OFTs show established but spatially impaired Crabp2 protein level asymmetry at E11.5, and its failure to progress to the medial region may be a secondary consequence of AP septation failure.

Together, these results show that *Ankrd11* ablation in the neural crest causes impaired BMP, mTORC1, and TGF-β signaling. Moreover, our results demonstrate that control OFTs exhibit asymmetric levels of expression of TGF-β and retinoic acid signaling pathway components and that this asymmetry is impaired in the *Ankrd11^ncko^* OFTs.

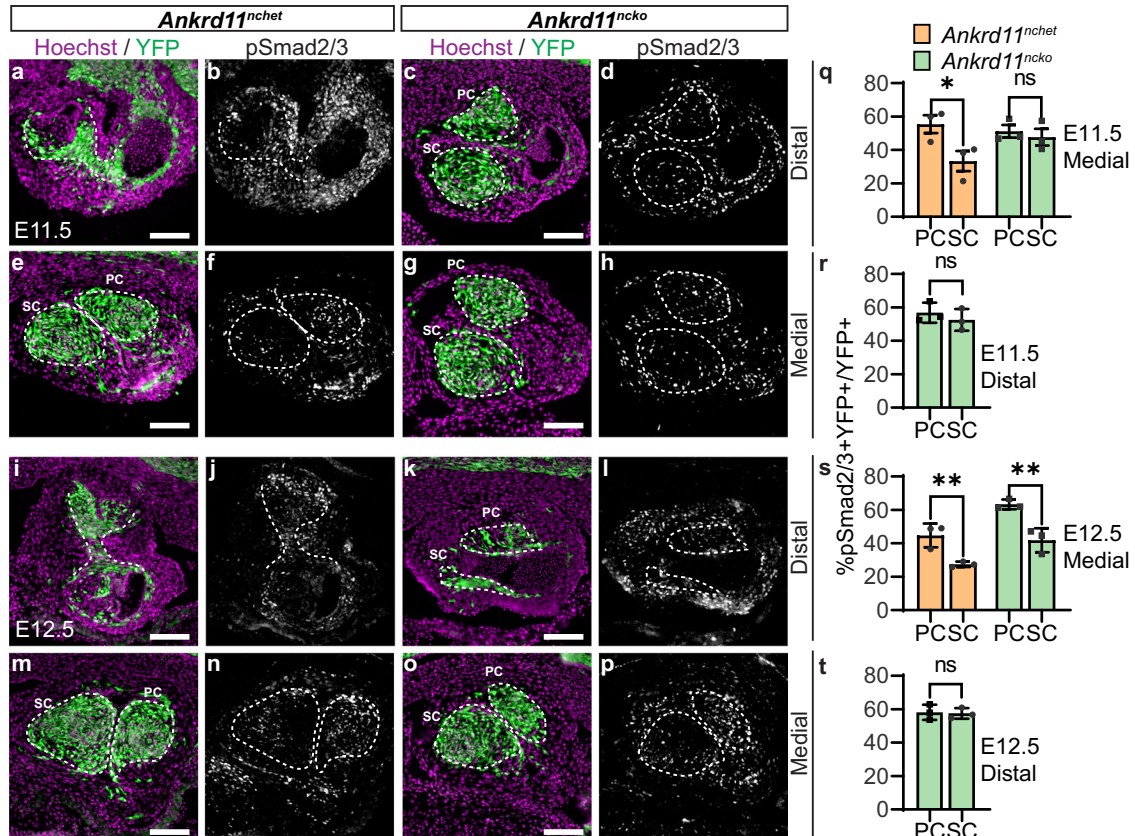

**Fig. 6 | Conditional loss of *Ankrd11* in the neural crest causes failure of asymmetric TGF-β signaling in OFT cushions at E11.5 that is partially restored by E12.5.** *Related to* Supplemental Fig. 4. **a–p** Representative images of E11.5 *Ankrd11*[nchet] (**a, b, e, f, i, j, m, n**) and *Ankrd11*[ncko] (**c, d, g, h, k, l, o, p**) distal and medial OFT, in the parietal (PC) and septal (SC) cushions (outlined with white dashed lines) at E11.5 (**a–h**) and E12.5 (**i–p**) immunostained for CNCCs (YFP; green), pSmad2/3 (white) and counterstained for nuclei (Hoechst, magenta). **q–t** Quantification of **a–p** for percent of pSmad2/3 CNCCs (% pSmad2/3 + YFP + /YFP+ cells) in

*Ankrd11*[nchet] (orange) and *Ankrd11*[ncko] (green) medial OFT (**q, s**) and in the *Ankrd11*[ncko] distal OFT (**r, t**), at E11.5 (**q, r**) and E12.5 (**s, t**), in the parietal and septal cushions. Two-tailed multiple t-tests with Holm-Sidak multiple comparisons test, medial (**q**, *Ankrd11*[nchet] $p = 0.030585$; **s**, *Ankrd11*[nchet] $p = 0.004133$, *Ankrd11*[ncko] $p = 0.002271$); two-tailed unpaired t-test, distal. Graphs represent mean ± s.e.m; $n = 3$ *Ankrd11*[nchet] and $n = 3$ *Ankrd11*[ncko] samples from at least 2 independent litters. ns: not significant, *$p < 0.05$, **$p < 0.01$. Scale bars: 100 μm. Source data are provided as a Source Data file.

## *Ankrd11*[ncko] CNCCs show a diverse set of dysregulated targets

We next sought to explore the changes to known effectors of OFT development using a high throughput method. Since we have shown that the embryonic OFT has a high spatiotemporal restriction of signaling pathway activation (Figs. 4–6 and S4), we decided to employ Multiplexed error-robust fluorescence in situ hybridization (MER-FISH), a single-cell spatial transcriptomic technique. We imaged 3 control and 3 *Ankrd11*[ncko] E11.5 anatomically matched embryo slices encompassing the medial OFT region using a custom panel of 140 RNA probes for genes (Supplementary Data 1) that are known to be important for OFT development and/or that were previously shown to be expressed in the OFT during the septation process in a scRNAseq dataset[4,7,8,36,39,47–52]. These include factors of important signaling pathways, extracellular matrix components, chemotactic ligands, transcription and chromatin remodeling factors, and cell type markers. Leiden analysis identified 10 clusters in the control and 13 clusters within the *Ankrd11*[ncko] OFT (Fig. 7a–c and Fig. S5). Clusters 11-13 contained a small number of cells that were only found in *Ankrd11*[ncko] OFTs (Fig. 7c). Clusters 2, 3, and 7 were identified to make up the OFT mesenchyme based on their location in the cushions and gene expression profiles (Fig. 7d–f). These three clusters showed high expression of the mesenchymal marker *Postn*, the mesenchymal and neural crest marker *Pdgfra*, and the neural crest marker *Sox9*[4,47,53,54]. Clusters 2 and 3 also expressed the marker *Penk* (Fig. 7d, f), which was previously identified as a marker only expressed by CNCCs of a more

mature phenotype that converge at the midline to form the AP septum[53]. Therefore, clusters 2, 3, and 7 were identified as CNCCs, although they may also include a small proportion of SHF-derived mesenchymal cells[31]. *Ankrd11*[ncko] OFTs had a smaller proportion of cells in cluster 2 and a larger proportion in cluster 3 when compared to control OFTs (Fig. 7e).

Differential gene expression analysis identified 40 differentially expressed genes (DEGs) in cluster 2, 28 DEGs in cluster 3, and 9 DEGs in cluster 7 (Fig. 8). In cluster 2, two of the downregulated DEGs in *Ankrd11*[ncko] cells were *Penk* and *Cxcl12* (Fig. 8a–c and Supplementary Data 2). As CNCCs progress through development, they undergo a transition from a mesenchymal to a vascular smooth muscle cell (VSMC) state to create the AP septum and line the arterial walls[47,53]. During this transition, the CNCCs upregulate their expression of *Cxcl12*, a chemokine encoding gene and a marker of immature VSMCs[47,53]. To this end, we observed the highest proportion of *Cxcl12* expression in cluster 2, with low proportions in clusters 3 and 7 of the control samples, in line with their expression of the maturity state marker *Penk*. In all three clusters, *Cxcl12* was downregulated in *Ankrd11*[ncko] cells (Fig. 8a–c and Supplementary Data 2). Conversely, we found upregulation of mesenchymal markers in *Ankrd11*[ncko] cells, including *Postn, Pdgfra*, and *Vim*[47,53–55], in all three clusters, and *Twist1* in clusters 2 and 7. Furthermore, there was downregulation of *Heyl*, a transcription factor downstream of the Notch pathway, and upregulation of *Tbx20*, a transcriptional repressor, in clusters 2 and 3 in

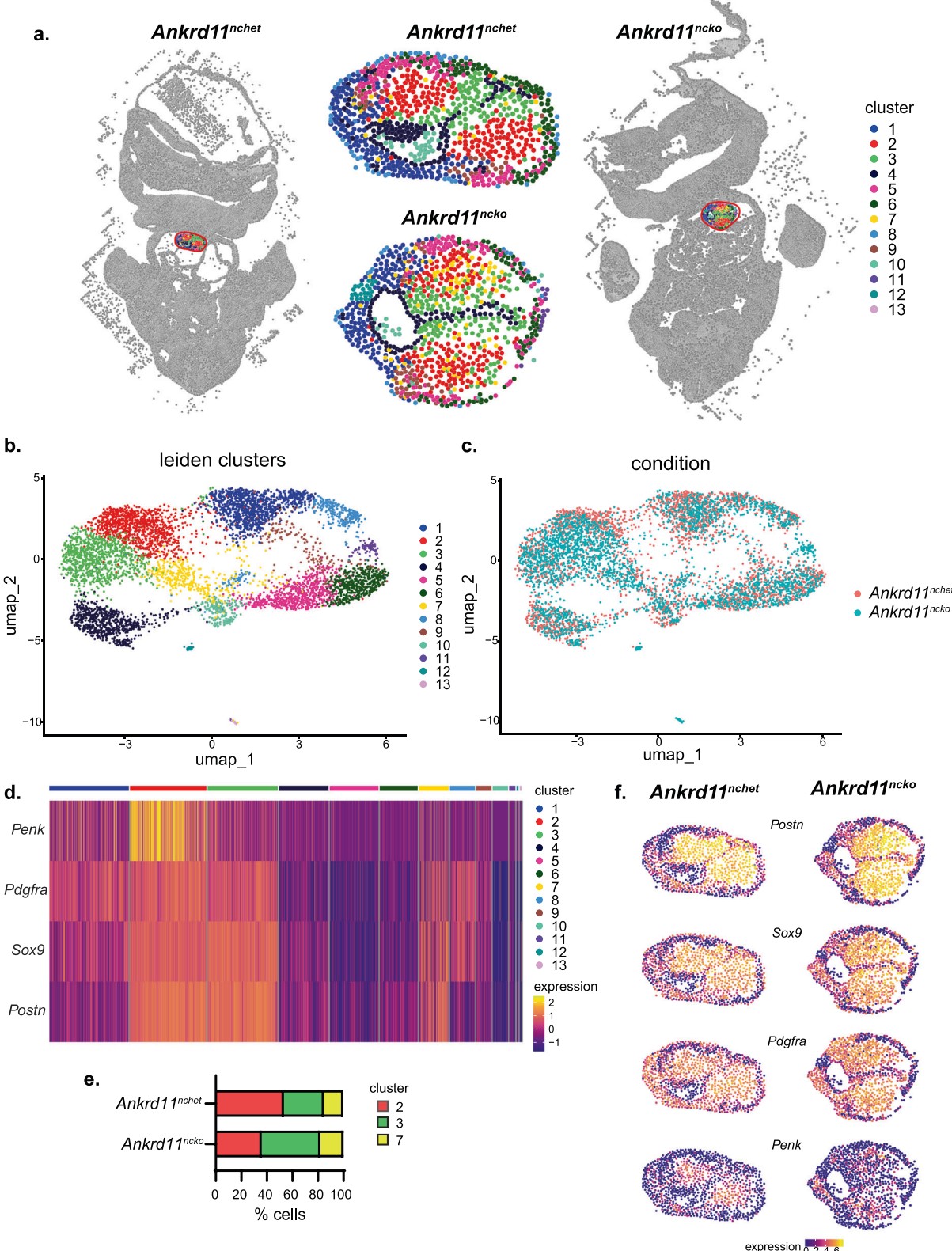

*Ankrd11[ncko]* cells (Fig. 8a–c and Supplementary Data 2)[47]. *Heyl* has been previously predicted as a positive regulator, and *Tbx20* as a negative regulator, of the mesenchymal to VSMC transition in the OFT[47]. These results suggest a delayed mesenchymal to VSMC transition of *Ankrd11[ncko]* CNCCs.

We observed DEGs in several signaling pathways known to be important for OFT development, including Wnt (*Fzd4, Vangl2, Fzd1,*

*Ctnnb1)*, BMP (*Id2, Bmp4, Bmpr2, Acvr1*), TGF-β (*Tgfb2*), Notch (*Heyl, Jag1, Notch1)*, Hippo (*Yap1*), JNK (*Jun*), Retinoic acid (*Rxra, Rarg*), FGF (*Fgfr2*), EGFR (*Erbb3*), and JAK-STAT (*Stat3*)[4,7,8,36,39,47–52] (Fig. 8a–c and Supplementary Data 2). We found downregulated expression of several growth factors and chemotactic ligands (*Sema3c, Efna5, Tgfb2*)[36,56–61]. In cluster 2, one of the most downregulated DEGs in *Ankrd11[ncko]* cells was *Sema3c*, which we have corroborated with

**Fig. 7 | MERFISH identified CNCC clusters in the OFT at E11.5.** *Related to* Supplemental Fig. 5. **a** Plot showing the spatial position of cells from representative transverse sections of E11.5 *Ankrd11nchet* (left) and *Ankrd11ncko* (right) embryos subjected to MERFISH with a custom 140 gene panel (Supplementary Data 1). A red bounding box defining the OFT is magnified and shown with cell clusters in the center. Cells in the OFT are colored according to Leiden clustering (clusters 1-13). **b**, **c** Plots showing the position of OFT cells from all samples in UMAP-space. Cells are colored according to Leiden clustering in **b** and according to genotype in **c**. **d** Heatmap plot showing scaled expression of *Penk, Pdgfra, Sox9,* and *Postn* in

OFT cells from all samples. Cells are ordered according to their clustering. **e** Bar plot showing the proportion of all CNC-classified cells belonging to different genotypes and clusters (*Ankrd11nchet*: cluster 2, 518 cells; cluster 3, 306 cells; cluster 7, 152 cells. *Ankrd11ncko*: cluster 2, 475 cells; cluster 3, 632 cells; cluster 7, 249 cells). **f** Plots showing spatial positions of OFT cells colored according to normalized gene expression. Left column shows cells from representative *Ankrd11nchet* sample, right column shows cells from representative *Ankrd11ncko* sample. n = 3 *Ankrd11nchet* and n = 3 *Ankrd11ncko* samples from at least 2 independent litters. Source data are provided as a Source Data file.

single-molecule FISH (RNAscope) (Fig. 8a–c, S6, Supplementary Data 2). Semaphorin 3c (Sema3c) is a chemoattractant and aggregation factor that plays a major role in CNCC migration into the OFT, their condensation in the OFT cushions, and their convergence to create the AP septum[36,56–59]. Furthermore, clusters 2 and 3 showed downregulation in neuropilin-1 (*Nrp1*), a known receptor for Sema3c[4], which may amplify the effects of the dysregulation. Other factors included ephrin A5 (*Efna5*) and transforming growth factor-beta 2 (*Tgfb2*), which were shown to control neural crest migration[60] and OFT morphogenesis[61], respectively (Fig. 8a–c and Supplementary Data 2). DEGs also included other transcription and chromatin remodeling factors important for OFT development such as *Egr1, Hoxa3, Ets1, Gata4, Sox11, Foxc1,* and *Chd7*[51,53,55], and cytoskeletal or extracellular matrix factors, such as *Cdh11, Mmp14, Adamts1,* and *Adam19*[62] (Fig. 8a–c and Supplementary Data 2). These results suggest that loss of Ankrd11 affects multiple signaling pathways, transcription factors, and chromatin remodelers important for CNCC function and OFT morphogenesis, further supporting and expanding our results in Figs. 4–6.

## Discussion

In this study, we demonstrate that Ankd11, a chromatin regulator and a causative gene for KBG syndrome, is a critical regulator of cardiac neural crest cell biology during embryonic heart development. Ablation of *Ankrd11* in the murine neural crest leads to dysregulation of CNCC organization and intracellular signaling, leading to aberrant OFT development and heart function.

We show that conditional knockout of *Ankrd11* in the neural crest leads to a failure of AP septation by CNCCs, creating a cardiac defect termed persistent truncus arteriosus (PTA) and an associated VSD. We also observed ventricular dilation without changes in wall thickness and decreased ventricular contractility. Ventricular dilation occurring without changes in wall thickness is indicative of eccentric ventricular hypertrophy, a compensatory mechanism to the dysfunctional ejection of blood from the ventricles, and consequent volume overload[63–65]. Ventricular dilation is associated with PTA in patients and animal models[65–69]. This can be observed as early as the fetal period due to truncal valve insufficiency, which causes volume overload through blood leakage from the truncal artery into the ventricles. Since we also observed truncal valve dysplasia in *Ankrd11ncko* hearts, this may indicate impaired valve function[70]. Notably, our results also show greater right ventricle dilation compared to the left ventricle. We postulate that this may be caused by the right ventricular origin of the truncal artery, which may force the right ventricle to be disproportionately impacted by the volume overload from a regurgitant truncal valve. Therefore, our results suggest that the ventricular dilation in *Ankrd11ncko* embryos is a compensatory mechanism at least in part due to PTA.

PTA in patients leads to congestive heart failure or cyanosis without surgical intervention[66,71]. Genetic models of PTA die *in utero* or perinatally, although the causes may be difficult to determine due to the multisystemic effects of their mutation[39,72–74]. While it was previously suggested that *Ankrd11ncko* neonates die due to cleft palate[28], our results suggest that a potentially fatal heart dysfunction may also contribute to their death.

At the cellular level, our results suggest that OFT septation failure in the *Ankrd11ncko* mice is primarily caused by impairment in CNCC organization within the OFT, and not by gross defects in CNCC migration into the OFT, proliferation, or survival. This differs from other genetic models of cardiac neural crest deficiency, where the cells are not able to infiltrate the most proximal regions of the OFT, indicating a severe migration defect[69]. Instead, we observed a delay in condensation in the *Ankrd11ncko* OFT cushions and failure to form the AP septum. These conclusions support and contrast other studies on Ankrd11. We have previously shown that *Ankrd11ncko* embryos display spatiotemporal impairment of cell organization and maturation of craniofacial neural crest-derived bone, suggesting that *Ankrd11* ablation in the neural crest causes the most dysregulation at later stages of craniofacial bone remodeling[28]. However, we also showed a decrease in cell proliferation in some, but not all, craniofacial regions, with no difference in apoptosis[28]. Moreover, our work with Yoda mice, which carry a heterozygous splice site-like mutation in *Ankrd11*, or with mice that have *Ankrd11* knocked down or out in neural stem cells, shows that Ankrd11 regulates embryonic neural stem cell proliferation[13,75]. We and others have also shown that Yoda mice or mice with *Ankrd11* knockdown in cortical progenitors and their derivatives have impaired neurogenesis and neuron positioning, which may suggest altered neuronal migration[13,76]. This was recently corroborated in our study, which showed that *Ankrd11* knockout in neural stem cells leads to severe neuroblast migration defect in the rostral migratory stream[75]. Together, the results presented in this study and previous reports suggest that Ankrd11 may have unique roles in each cell type it is expressed in and that Ankrd11's function may depend on cellular identity and environment.

*Ankrd11ncko* CNCCs show dysregulation of several signaling pathways, which may contribute to the observed delay in condensation and failure to form the AP septum. The TGF-β superfamily plays an important role in OFT development. Neural crest conditional knockout of TGF-β receptors, including transforming growth factor-beta receptor type 1 (*Tgfbr1*) and type 2 (*Tgfbr2*) results in PTA with interrupted aortic arch as well as varying effects on cell number, apoptosis, and smooth muscle formation[42,77]. Of the three TGF-β ligands, only *Tgfb2* global knockout results in cardiac defects including VSD and double outlet right ventricle[61]. Ablation of downstream factors *Smad4* and *Smad7* that are involved in the Smad2/3-dependent TGF-β and Smad1/ 5/8-dependent BMP pathways also causes OFT defects[45,78,79]. While *Tgfb2* expression was downregulated in *Ankrd11ncko* CNCCs in MERFISH results, it is unclear how this contributes to pSmad2/3 signaling abnormalities in the *Ankrd11ncko* CNCCs. Here, we have identified asymmetric pSmad2/3 signaling by CNCCs in control medial OFTs. Notably, pSmad2/3 expression was symmetric in E11.5 and asymmetric in E12.5 medial, but not distal, *Ankrd11ncko* OFT. Since we did not analyze control distal OFT cushions just prior to septation, it is unknown whether pSmad2/3 asymmetry is normally established distally.

Since *Ankrd11ncko* CNCCs were able to establish an asymmetric distribution of Crabp2 and not of pSmad2/3 in distal E11.5 OFT, this suggests that loss of Ankrd11 in the neural crest impairs specific asymmetrically activated signaling pathways, rather than impairing all asymmetry within the CNCC population. Notably, as MERFISH results identified dysregulated expression of the *Rxra* and *Rarg*

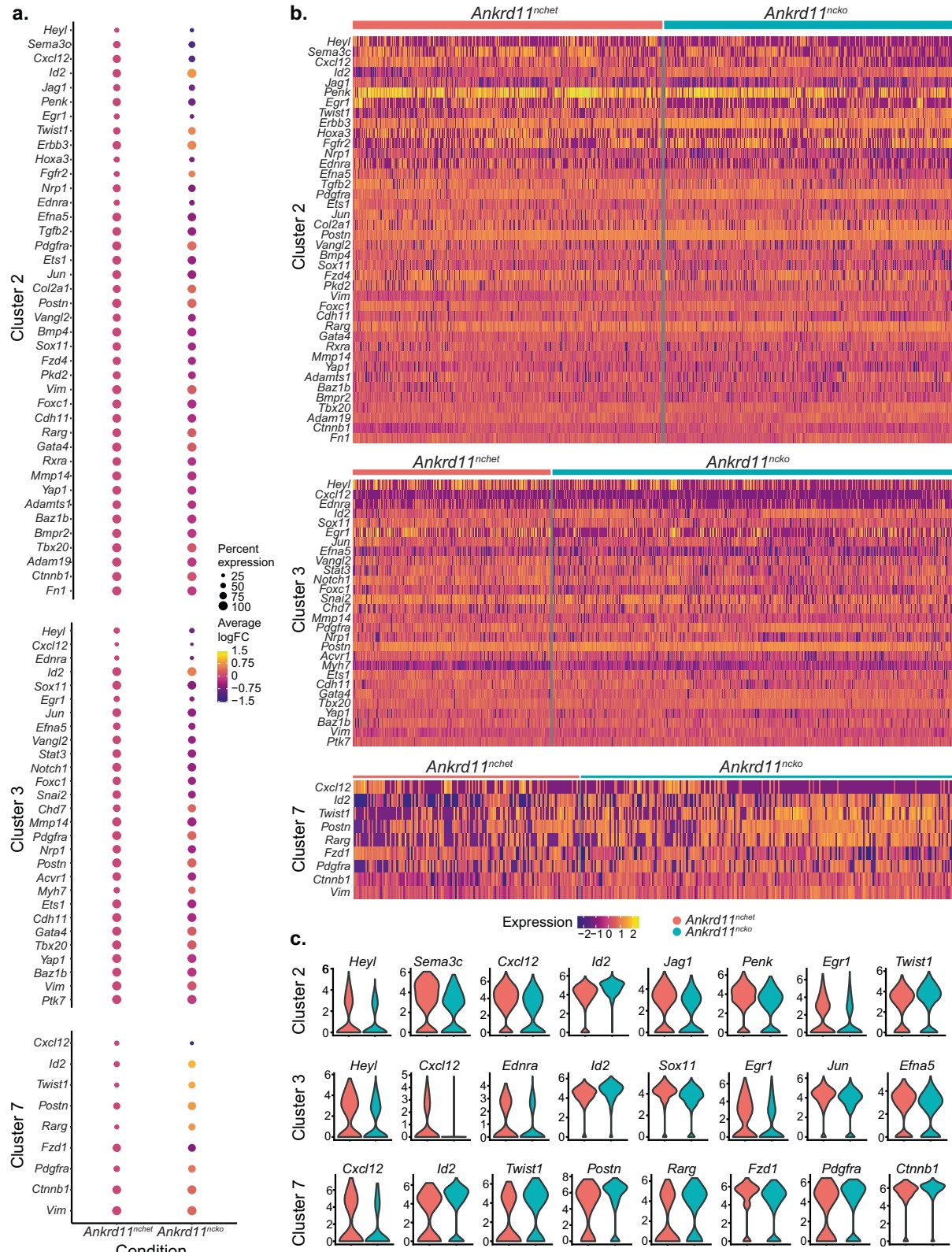

receptors in the *Ankrd11^{ncko}* OFT, this suggests that retinoic acid signaling may be dysregulated through a different mechanism in *Ankrd11^{ncko}* CNCCs.

However, nothing is known about the mechanism behind the pSmad2/3 and Crabp2 asymmetry within CNCCs or its downstream effects on AP septum formation. Left-right asymmetry is crucial for proper embryo development, and several factors are known to contribute to cardiac left-right asymmetry, including the TGF-β superfamily[4]. Activin-like TGF-β receptors *Acvr2b* and *CFC1* and ligand *Gdf1* are involved in left-right patterning in heart development[80–82], and global knockouts of these genes lead to cardiac defects such as transposition of the great arteries. Our results show

**Fig. 8 | Conditional loss of *Ankrd11* in the neural crest causes dysregulated expression of a diverse set of genes at E11.5.** *Related to* Supplemental Fig. 6. **a** Dotplots showing statistically significantly differentially expressed genes (DEGs) between the *Ankrd11^nchet^* and *Ankrd11^ncko^* genotypes for each cluster. Dots are colored according to average log fold-change of expression from *Ankrd11^nchet^* genotype (negative change indicates expression is lower in *Ankrd11^ncko^* compared to *Ankrd11^nchet^*, and a positive change indicates higher expression). The dot radius corresponds to the percent of cells in each cluster that express (i.e., show the presence of at least one transcript) the gene of interest. Genes displayed are ordered according to absolute average log2 fold-change (highest at the top, lowest

at the bottom). **b** Heatmap plot showing scaled expression of differentially expressed genes between the *Ankrd11^nchet^* and *Ankrd11^ncko^* genotypes for each cluster. Cells are ordered according to their clustering. Genes displayed are ordered according to absolute average log2 fold-change (highest at the top, lowest at the bottom). **c** Violin plots showing scaled expression of selected differentially expressed genes between the *Ankrd11^nchet^* and *Ankrd11^ncko^* for each cluster. Genes were selected by taking the 8 highest absolute average log2 fold-change values from each marker group. $n = 3$ *Ankrd11^nchet^* and $n = 3$ *Ankrd11^ncko^* samples from at least 2 independent litters. Source data are provided as a Source Data file.

further evidence that the TGF-β pathway contributes to OFT asymmetry during OFT septation. There is also evidence of left-right asymmetry within the CNCC population. Gandhi et al.[83] observed that ablation of the right cardiac neural folds in chick, which removes CNCCs from the right side of the embryo, produced a more severe septation impairment compared to ablation of the left cardiac neural folds.

A subfamily within the TGF-β superfamily, the BMP pathway is a well-known regulator of cardiac neural crest development, with manipulation of its different components also leading to variable phenotypes[4]. Neural crest-specific ablation of the Activin-type BMP receptor *Acvr1* causes decreased CNCC migration into the OFT, failure of OFT septation, and impaired smooth muscle formation[69], neural crest-specific heterozygous deletion of the BMP receptor *Bmpr2* causes an overriding aorta[84], while neural crest-specific deletion of *Bmpr1a* leads to septation failure with normal CNCC migration[74]. The downregulation of *Bmp4, Bmpr2,* and *Acvr1* that was found in *Ankrd11^ncko^* MERFISH results may contribute to the decreased pSmad1/5/8 signaling.

Neural crest-specific ablation of *Ctdnep1*, a BMP inhibitor, causes an opposing phenotype to the defects observed in the *Ankrd11^ncko^* embryos, specifically premature CNCC condensation and asymmetric AP septum formation, as well as increased pSmad1/5/8 activity[36]. Furthermore, the BMP pathway was found to at least in part regulate *Sema3c* expression[36], a crucial factor for CNCC condensation that was downregulated in the *Ankrd11^ncko^* OFT in MERFISH and RNAscope results. Sema3c is a glycoprotein that is secreted within the OFT and pharyngeal arches, and is considered a chemoattractant guidance signal to promote CNCC migration into these structures, their condensation within the cardiac cushions and their migration to create the AP septum[36,56–59]. Overactivation of BMP signaling in CNCCs causes increased Sema3c expression, contributing to an overcondensation phenotype[36]. Our findings support these results by showing that a reduction in BMP signaling in *Ankrd11^ncko^* OFT mesenchyme at E11.5 correlates with delayed CNCC condensation since both the pSmad1/5/8 signal and condensation were restored at E12.5. Together, this provides evidence that the fine-tuning of BMP signaling is important for OFT condensation.

Recent studies have shown that mTOR signaling contributes to neural crest development. *mTOR* deletion causes decreased CNCC migration, proliferation, and differentiation, leading to PTA[39]. *mTOR* deletion in both cranial and cardiac neural crest causes decreased F-actin levels and consequent defects in cytoskeletal dynamics, which were attributed to mTORC1 pathway dysregulation[39,85]. mTORC1 is known to regulate cell migration in multiple contexts, with functions in cytoskeletal and extracellular matrix remodeling as well as focal adhesion formation[86–88]. Notably, *Ankrd11* knockdown in murine embryonic cortical neuron cultures was previously shown to reduce pS6 levels, thus implicating mTORC1 signaling[76].

Together, our study confirms mTORC1 and identifies TGF-β and BMP signaling pathways to be affected during OFT septation by the loss of Ankrd11. Furthermore, our MERFISH results identified other pathways that may be dysregulated in the *Ankrd11^ncko^* OFT, including Wnt, Notch, and Retinoic acid pathways.

From a clinical perspective, cardiac defects are detected in up to 40% of KBG syndrome patients[23,25,26]. Moreover, *ANKRD11* variants are reported in large-scale exome and genome sequencing studies of gene-disease associations in CHD, including conotruncal defects[89–92]. The reported defects include aortic coarctation, patent ductus arteriosus, valve dysplasia, and atrial and ventricular septal defects[23,25,26]. Such defects are also found in many other neural crest-mediated developmental disorders, including DiGeorge and Noonan syndromes[5]. Our results demonstrate a critical and direct role for ANKRD11 in regulating cardiac neural crest-mediated heart development and function. While our work has not tested the role of ANKRD11 in non-neural crest-derived heart tissues, our results suggest neural crest involvement in KBG syndrome cardiac defects.

## Methods
### Animal models

All animal use was approved by the Animal Care Committee of the University of Alberta in accordance with the Canadian Council of Animal Care policies. All mice were housed in a University of Alberta Animal Facility and serviced by Health Sciences Laboratory Animal Services (HSLAS). Mice were maintained on a 14/10 h light/dark cycle, at 21–23 °C room temperature, and 40–70% humidity. Food and water were provided ad libitum. Embryos at developmental ages between E10.5-18.5 and of either sex were used for all experiments.

*Ankrd11^fl/fl^* mice, where exon 7 of the *Ankrd11* gene was flanked by LoxP sites, were derived from the *Ankrd11^TM1a(EUCOMM)Wtsi/IcsOrl^* (*Ankrd11^tm1a^*) sperm (EM:07651, the European Mouse Mutant Archive–Infrafrontier) as described in ref. 28. Ankrd11^fl/fl^ mice were bred to hemizygous *Wnt1Cre2+* mice (B6.Cg-E2f1Tg(Wnt1-cre)2Sor/J, stock # 022501, Jackson Laboratories)[30], to create the *Ankrd11^fl/fl^; Wnt1Cre*[2] mice. These were used for μCT analysis. Neural crest cell lineage tracing was performed by breeding in the *RosaYFP^STOP^* allele using the *RosaYFP^STOP/STOP^* mice (B6.129X1-Gt(ROSA)26Sortm1(EYFP)Cos/J, stock # 006148, Jackson Laboratories)[93]. These mice were used for all other analysis. The mouse line was maintained by crossing *Ankrd11^WT/WT^;RosaYFP^STOP/STOP^;WntCre*[2] with *Ankrd11^fl/fl^; RosaYFP^STOP/STOP^* mice. Resulting *Ankrd11^fl/WT^;RosaYFP^STOP/STOP^;WntCre*[2] were timed mated with *Ankrd11^fl/fl^;RosaYFP^STOP/STOP^* or *Ankrd11^fl/WT^; RosaYFP^STOP/STOP^* mice to generate *Ankrd11^WT/WT^;RosaYFP^STOP/STOP^; Wnt1Cre^2+^* (*Ankrd11^WT/WT^*), *Ankrd11^fl/fl^;RosaYFP^STOP/STOP^* (*Ankrd11^fl/fl^*) and *Ankrd11^fl/WT^;RosaYFP^STOP/STOP^* (*Ankrd11^fl/WT^*), *Ankrd11^fl/WT^;Rosa YFP^STOP/STOP^;Wnt1Cre*[2] (*Ankrd11^nchet^*) as well as *Ankrd11^fl/fl^;RosaYFP^STOP/STOP^; Wnt1Cre*[2] (*Ankrd11^ncko^*). Mice were mated in the evenings and a plug was determined in the morning, with a positive plug considered as E0.25.

All mice were bred and genotyped using the following primers: *Ankrd11:* forward, 5′-CTGTCTCAGAGAGGAGAGTGAGGAGGAC-3′; reverse, 3′- TACCTTACACCCTGAGACGGCGTC-5′, 34 cycles of 94 °C–30 s, 62 °C–45 s, 72 °C–60 s. Pan-Cre: forward, 5′-TTCCCGCAGAACCTGAAGATG -3′; reverse, 3′- CCCCAGAAATGCCAGATTACG-5′; control forward primer 5′-AACAACAATGGCACAACCTAAT-3′; control reverses 3′-ACTTTCTCCCCACCCGTCTA-5′; 35 cycles of 94 °C–15 s, 60 °C–30 s, 72 °C–90 s. YFP^STOP^: wild-type, 5′-GGAGCGGGAGAAATGGATATG-3′; common, 5′- AAAGTCGCTCTGAGTTGTTAT-3′;

mutant, 5′-AAGACCGCGAAGAGTTTGTC-3′, 35 cycles of 94 °C–15 s, 65 °C–15 s, 72 °C–60 s.

### Reagents

**Primary antibodies and stains.** Anti-Crabp2 (Proteintech, 10225-1-AP, Lot 00051203, 1:200), anti-eGFP (Abcam, ab13970, Lot GR3361051-16, 1:2000), anti-Ki67 (BD Pharmingen, 556003, Lot 1119219, 1:500), anti-αSMA (Sigma, A2547, Lot 099M4848V, 1:1000), anti-PDGFRα (R&D Systems, AF1062, Lot HMQ0220101, 1:500), anti-pSmad1/5/8 (Cell Signaling, 13820, Lot 4, 1:300), anti-pSmad2/3 (Cell Signaling, 8828, Lot 8, 1:300), anti-S6 (Cell Signaling, 2317, Lot 13, 1:600), anti-pS6 (phosphoSer240/244; Cell Signaling, 2215, Lot 18, 1:600), biotinylated IB4 (Vector Laboratories, VECTB1205, Lot ZB1017, 1:1000), Phalloidin-iFluor647 (Abcam, ab176759, Lot GR3279773-9, 1:1000). If the antibody was of mouse origin, then a M.O.M. Kit (Vector Biolabs) with appropriate streptavidin incubation (please see information below) was used according to instructions.

**Secondary antibodies.** Anti-chicken-Alexa488 (Jackson, 703-545-155, Lot 162189, 1:1000), anti-chicken-Alexa647 (Jackson, 703-605-155, Lot 153969, 1:1000), anti-goat-Alexa647 (Jackson, 705-605-147, Lot 154191, 1:1000), anti-rabbit-Alexa647 (Jackson, 711-605-152, Lot 154880, 1:1000), streptavidin-Cy3 (Jackson, 016-160-084, Lot 168280(2), 1:1000), streptavidin-Cy5 (Jackson, 016-170-084, Lot 151872, 1:1000).

### Micro-computed tomography (μCT)

Embryos were dissected at E18.5 and fixed overnight in 4% PFA, followed by 48 h in Lugol's iodine solution (Electron Microscopy Sciences, 26658-04), as per[94]. Embryos were imaged using a Milabs UHT-μCT scanner (Milabs, Utrecht, The Netherlands) of the School of Dentistry using the following parameters: step angle: 0.1 degrees, voltage: 50 kV, current: 0.24 mA, exposure time: 75 ms. Scans were exported as NII files, reconstructed with 15 μm voxel size, and analyzed using 3D Slicer (https://www.slicer.org/)[95] (version 5.0.3) for ventricular volumes and wall thicknesses.

### In utero echocardiography

Fetal cardiac structure and function were assessed by echocardiography using a 32 MHz linear array transducer (Vevo 3100, Visualsonics, Canada). Dams pregnant with E18.5 embryos were sedated with 2.5% isoflurane and maintained at 1.25–1.5% isoflurane using 0.8 L/min of oxygen. The abdomen was shaved and treated with depilatory cream. In a supine position, limbs were fixed to electrodes with tape and conductive electrode gel (SigmaGel, Parker Labs), and electrocardiogram, respiratory rate, and body temperature were continuously recorded. Body temperature was maintained with a heated table surface and heat lamp, and a heated ultrasound transmission gel was used. The total duration of all procedures did not exceed 1 h per dam and was performed by a single operator.

Each fetus was hand scanned using B-mode images to identify basic cardiac structure; fetuses with evident truncus arteriosus and ventricle septal defects (VSD) were preliminarily identified as *Ankrd11^{ncko}*. Color Doppler images were then used to confirm VSD and truncus arteriosus. The area of the ventricular chambers was measured along the short and long-axis of the fetus using B-Mode images. Changes between the ventricular end-diastolic area and end-systolic area were used to calculate fractional area change (FAC) in the right and left ventricles. Using M-mode images in the short axis of the fetus, ventricular wall thicknesses and chamber diameters were measured in end-systole and end-diastole. Changes between the end-diastolic and end-systolic ventricular diameters were used to calculate fractional shortening. Finally, heart rate was calculated by counting heartbeats over two separate 10 s B-mode images. After imaging, the dam was euthanized with $CO_2$, and fetuses were dissected to confirm the identification of the *Ankrd11^{ncko}* and control (*Ankrd11^{fl/fl}*, *Ankrd11^{fl/WT}* and *Ankrd11^{nchet}*) embryos.

### Immunohistochemistry

**Tissue preparation.** Pregnant dams were euthanized with $CO_2$ when embryos were at E10.5, E11.5, E12.5 and 18.5. Whole embryos (E10.5-E12.5) or dissected hearts (E18.5) were fixed in 4% paraformaldehyde (PFA; Electron Microscopy Sciences, 15710) for 24 h at 4 °C followed by three changes of 30% sucrose in 1xPBS every 24 h. Samples were embedded and flash frozen in an optimal cutting temperature (OCT) compound (ThermoFisher) and sectioned on a cryotome (CM1950, Leica Biosystems, Germany) to 14 μm slices for IHC, BaseScope or RNAscope assays and at 10 μm for MERFISH assay. The slices were transferred onto glass microscope slides and stored at –80 °C for IHC, BaseScope, or RNAscope assays or dehydrated with ethanol as per Vizgen® MERFISH manufacturer's instructions and stored at 4 °C for a maximum 1 week.

**Immunofluorescence and apoptosis detection.** Sections were processed as described in[96–98]. Briefly, sections were dried at 37 °C, washed with phosphate-buffered saline (PBS), and blocked for 1 h (5% bovine serum albumin [BSA, Jackson ImmunoResearch, 001-000-162], 0.3% Triton-X100 in PBS), then incubated with primary antibodies listed in "Reagents" section (in 5% BSA) overnight at 4 °C. Sections were incubated with secondary antibodies listed in the "Reagents" section for 1 h at room temperature. If the antibody was of mouse origin, then a Vector Laboratories M.O.M. Kit was used according to instructions. Samples were counterstained with the nuclear stain Hoechst (33217; Riodel-De Haen Ag) at 1:1000 and mounted with Fluoromount-G mounting media (ThermoFisher, 00-4958-02).

Apoptotic cells were detected through the TUNEL (Terminal deoxynucleotidyl transferase dUTP nick end labeling; ApopTag Red In Situ Apoptosis Detection Kit, Millipore, S7165) assay according to the manufacturer's protocol with modifications. After immunohistochemistry, sections were post-fixed in 4% PFA for 10 min at room temperature, followed by 5 min incubation in 2:1 ethanol:acetic acid at –20 °C. After washing, 50 μL per slide of equilibration buffer was applied for 10 s, followed by 100 μL of working strength TdT (terminal deoxynucleotidyl transferase) enzyme for 1 h at 37 °C. Slides were washed with 400 μL of stop/wash buffer for 10 min and 100 μL of anti-digoxigenin/rhodamine conjugate was applied for 30 min. Samples were counterstained with the nuclear stain Hoechst at 1:1000 and mounted with Fluoromount-G mounting media.

### RNAscope and BaseScope

RNAscope was performed according to refs. 96,98 and manufacturer's protocol with modifications (RNAscope Multiplex Fluorescent Reagent Kit, ACDBio). Briefly, sections were dried at 37 °C, dehydrated with ethanol, rehydrated with PBS, and incubated in 15 μl of 10% Protease IV (322340) diluted in PBS for 10 min at 37 °C. Sections were then washed and incubated with mRNA probes for 2 h at 40 °C, followed by consecutive incubations with AMP-1 to 3 (320852-320854) and AMP-4B (320856) or AMP-4C (320857). We used the *Sema3c-C3* probe (441441-C3) and negative control provided by the manufacturer (320871). After RNAScope, samples were blocked for 1 h (5% BSA, 0.3% Triton-X100 in PBS), then incubated with primary antibodies listed in "Reagents" section (in 5% BSA) overnight at 4 °C. Sections were incubated with secondary antibodies listed in the "Reagents" section for 1 h at room temperature. Samples were counterstained with the nuclear stain Hoechst at 1:1000 and mounted with Fluoromount-G mounting media.

BaseScope was performed according to the manufacturer's protocol with modifications (BaseScope v2 Kit-Red, ACDBio). Sections were taken directly from –80 °C and washed with 1x PBS 2 times; sections were left in the 3rd wash for 5 min, shaking gently. Sections

were baked at 60 °C for 30 min and post-fixed with 4% paraformaldehyde (15710, Fisher) at 4 °C for 15 min. Sections were dehydrated for 5 min with 50% and 70% ethanol, and 2 × 5 min with 100% ethanol at RT. Ethanol was evaporated, and sections were treated with hydrogen peroxide (322381, ACDBio) for 10 min at RT. Sections were washed once with water, then boiled for 5 min in 1x Target Retrieval (322000, ACDBio). Sections were washed twice with water, shaking gently. Sections were treated with 100% ethanol for 5 min and dried at RT overnight. Samples were treated with 1:3 Protease III (322337, ACDBio) diluted 1x PBS at 40 °C for 25 min. Sections were washed twice with water.

Sections were incubated with either the probe *Ankrd11-C1* (1091241-C1, ACDBio), positive or negative control probes supplied with the kit for 2 h at 40 °C. Slides were washed 2 × 2 min with 1X RNAScope Wash Buffer (310091, ACDBio), followed by consecutive incubations with AMP-1 to 8; Amps 1-2, 4, 5 were incubated for 30 min at 40 °C, Amps 3 and 6 were incubated for 15 min at 40 °C, Amp 7 was incubated for 30 min at RT, and Amp 8 was incubated for 15 min at RT (323911-323918, ACDBio). Sections were washed 2 × 2 min with 1x RNAScope Wash buffer in between each Amp incubation. FastRed solution (1:60 Fast B: Fast A, 322918, ACDBio) was used to detect the signal; FastRed solution was added to sections for 10 min at RT followed by 2 × 2 min 1x PBS wash. Samples were counterstained with nuclear stain DAPI (320858, ACDBio), washed for 3 × 5 min with 1x PBS, and mounted with Fluoromount-G mounting media.

## MERFISH

**Probe design and sample preparation.** A total of 140 genes were identified and evaluated using the MERSCOPE® Gene Panel Design Portal available at Vizgen (portal.vizgen.com) to ensure that each gene was suitable in length for probe binding and that abundance was below the recommended threshold to avoid optical crowding during imaging.

Paraformaldehyde-fixed (PFA) frozen mouse embryos were embedded in optimal cutting temperature (OCT) compound and sectioned into 10 μm thick sections on a cryostat at −20 °C for placement on fiducial bead-coated MERSCOPE Slides (Vizgen 20400001). The tissue slices were washed three times with 5 mL 1x PBS and incubated with 70% ethanol at 4 °C overnight for tissue permeabilization.

**Antibody staining and probe hybridization.** Cell boundary staining was performed using Vizgen's Cell Boundary Staining Kit (Vizgen 10400009) per the user guide for fresh- and fixed-frozen sample preparation. Briefly, samples were washed with 5 mL 1x PBS and then blocked for 1 h at room temperature in Blocking Buffer C Premix (Vizgen 20300100) with RNase inhibitor (New England Biolabs M0314L) added at 1:20 dilution. For primary antibody staining, samples were incubated for 1 h at room temperature with Cell Boundary Primary Stain Mix (Vizgen 20300010) at 1:100 dilution and RNase inhibitor (New England Biolabs M0314L) at 1:20 dilution in blocking buffer, followed by three washes with 5 mL 1x PBS. Samples were then incubated for 1 h at room temperature with Cell Boundary Secondary Stain Mix (Vizgen 20300011) at 1:33 dilution and RNase inhibitor (New England Biolabs M0314L) at 1:20 dilution in blocking buffer, then washed three times with 5 mL 1x PBS, post-fixed in 5 mL 4% paraformaldehyde in 1x PBS for 15 min at room temperature, and washed twice with 5 mL 1x PBS. Samples were then incubated in Formamide Wash Buffer (Vizgen 20300002) for 30 min at 37 °C followed by probe hybridization with 50 μL of a custom-designed MERSCOPE Gene Panel Mix (Supplementary Data 1) at 37 °C for 36–48 h.

**Gel embedding and tissue clearing.** Following incubation, the tissues were washed twice with 5 mL Formamide Wash Buffer (Vizgen 20300002) at 47 °C for 30 min, and embedded into a hydrogel using the Gel Embedding Premix (Vizgen 20300004), ammonium

persulfate (Sigma, 09913-100G) and TEMED (N,N,N',N'-tetramethylethylenediamine) (Sigma, T7024-25ML) from the MERSCOPE Sample Prep Kit (Vizgen 10400012). After a 1.5-h incubation at room temperature, the samples were cleared with a solution consisting of 50 μl of Proteinase K (NEB, P8107S) and 5 mL of Clearing Premix (Vizgen 20300003) at 37 °C for at least 24 h, or until the tissue became transparent.

**Imaging.** After removing the clearing solution and washing three times with Sample Prep Wash Buffer (Vizgen 20300001), the samples were stained with DAPI and Poly T Reagent (Vizgen 20300021) for 15 min at room temperature, washed for 10 min with 5 mL of Formamide Wash Buffer (Vizgen 20300002), then washed for 5 min with 5 mL of Sample Prep Wash Buffer (Vizgen 20300001). The imaging reagents and processed sample were loaded into the MERSCOPE and a low-resolution DAPI mosaic (10x magnification) was used to select the region of interest before high-resolution imaging on the MERSCOPE system (Vizgen 10000001). With a 60x oil immersion objective, DAPI and Poly T stains were imaged at 7 focal planes on the z-axis for each tiled field of view (FOV), followed by 6 rounds of 3-color imaging across all focal planes using 750 nm, 650 nm, and 560 nm laser illumination. Each imaging round was followed by incubation in extinguishing, rinse, hybridization, wash, and imaging buffers. Additionally, a single image of fiducial beads was acquired at each FOV using 488 nm illumination. The full instrumentation protocol is available at https://vizgen.com/resources/merscope-instrument/.

**Image analysis.** The MERlin image analysis pipeline v0.1.6[99] was used to analyze the raw image files from the MERSCOPE experiment to align image stacks from the different MERFISH rounds, filter out background noise, and enhance RNA spot detection. Individual RNA molecule barcodes were then decoded using a pixel-based algorithm and an adaptive barcoding scheme that corrects misidentified barcodes not matching the provided codebook. Next, cells were segmented using information from nuclear DAPI and cell membrane antibody stains with the Cellpose software package[100]. The decoded RNA molecules were then grouped into individual cells, generating single-cell RNA count matrices. The entire pipeline was run for each imaging FOV and then tiled over the entire sample imaging area, or up to 1 cm$^2$.

Four samples (n = 2 *Ankrd11*$^{nchet}$ and n = 2 *Ankrd11*$^{ncko}$) were processed in the Vizgen facility, and 2 samples (additional n = 1 *Ankrd11*$^{nchet}$ and n = 1 *Ankrd11*$^{ncko}$) were processed at the University of Alberta, Advanced Cell Exploration Core facility. Altogether, n = 3 samples per genotype from 2-3 litters were processed and analyzed.

## Microscopy
Phalloidin-stained as well as RNAscope and BaseScope sections were imaged with a Zeiss LSM700 confocal microscope with a photomultiplier tube (PMT) using the 40x objective. Z-stacks spanning ~10–14 μm were captured with an optical slice thickness of 0.85-1.5 μm. All other images were captured using a Zeiss Axio Imager M2 fluorescence microscope with ORCA-Flash LT sCMOS Camera using the 20x objective for OFT and heart sections and the 2.5x objective for whole embryos in a single plane. All image acquisition was performed using Zen software (Zeiss). Confocal images were processed into stacked Z-planes. The thymus was imaged with a dissection microscope using a Samsung A53 phone camera.

## Image analysis
To analyze CNCC-derived smooth muscle cells in the arteries at E18.5, 1 proximal and 1 distal image was used from each heart. αSMA + YFP+ cells and total αSMA+ cells were counted within the αSMA+ ring of cells surrounding the artery lumen. For *Ankrd11*$^{nchet}$,

cell numbers represent the combined numbers from the aorta and pulmonary trunk.

To analyze CNCC number, proliferation, and apoptosis, at least 6 transverse OFT sections from E10.5 embryos were split into proximal, medial, and distal regions using morphological characteristics. At E10.5, the proximal region has a small lumen and dispersed CNCCs. The medial region has a large lumen, dispersed CNCCs, and a thickened bottom cushion. The distal region has a large lumen, more compacted CNCCs, and a thickened bottom cushion. YFP+ CNCCs were counted and averaged to show the mean average number of cells per OFT section in each region. The proliferative index is presented as % Ki67+YFP+ cells over total YFP+ cells. The apoptotic index is presented as % TUNEL+YFP+ cells over total YFP+ cells.

To analyze anatomically matched sections of $Ankrd11^{nchet}$ and $Ankrd11^{ncko}$ OFTs at E11.5 and E12.5, the $Ankrd11^{ncko}$ distal region was determined to be the equivalent region to the $Ankrd11^{nchet}$ septated region based on the number of OFT sections it occupied, and the medial sections used for analysis were adjacent to this boundary. This method was used to avoid relying on the differing morphology between $Ankrd11^{nchet}$ and $Ankrd11^{ncko}$ OFTs and between embryonic ages.

To determine CNCC nuclear orientation in the OFT, all YFP+ cells from three sections of the medial region per OFT were analyzed as per ref. 36 with slight modifications. Namely, the angle between the long-axis of the CNCC nucleus and the OFT cushion center was manually measured using the Angle tool in Fiji ImageJ and binned into the 90°-70°, 69°-50°, 49°-30°, 29°-0° categories. The OFT cushion center was visually identified as a cluster of CNCCs that acted as a focal point around which the rest of the CNCCs was organized in approximate spirals.

To investigate $Sema3c$ mRNA expression via RNAscope, images from at least 3 medial sections were analyzed from each embryo, and due to the varying distal-proximal gradient of marker expression, sections with the highest level of expression were quantified. Cells with three or more RNAscope dots were considered positive for marker expression as described in refs. 101,102. In $Ankrd11^{nchet}$ OFT, % $Sema3c$+YFP+ over total YFP+ cells were graphed from one section closest to the AP septum. In $Ankrd11^{ncko}$ OFT, 3 medial sections were quantified, and one section with the highest % $Sema3c$+YFP+ over total YFP+ cells was graphed. An average of ~400 cells was counted per OFT section.

To investigate $Ankrd11$ mRNA expression via BaseScope, images from 1 medial section were analyzed from each embryo. Within the OFT mesenchyme, DAPI+ cells with at least 1 $Ankrd11$ probe count/cell were counted as $Ankrd11$+. Additionally for $Ankrd11^{nchet}$, $Ankrd11$+ cells were stratified by the number of $Ankrd11$ probe counts.

To analyze pS6 activation in the OFT, the % pS6 + S6 + YFP+ over total S6 + YFP+ cells from two sections was averaged to show the mean percentage of cells per medial OFT section. To analyze pSmad1/5/8, pSmad2/3, and Crabp2 signal, a threshold was manually set to delineate cells with high marker signal and cells with low marker signal, and only YFP+ cells with high marker signal were counted and presented as % marker+YFP+/YFP+ cells. An average of ~400 cells was counted per OFT section. Two sections were counted per OFT region.

YFP+marker+ CNCCs were counted when YFP+ cell cytoplasm overlayed with a Hoechst+ nucleus and a marker+ signal. All cells were counted manually.

To analyze the E18.5 thymus, both lobes of the thymus gland were outlined by hand and the area was quantified using a reference scale.

Images were counted and analyzed for fluorescence intensity, and representative images were processed in Fiji ImageJ v1.53t software[103]. Figures were generated in Adobe Illustrator CC 2015.

### Single-cell RNA sequencing data re-analysis

The mouse trunk NC datasets at E9.5 and E10.5, published in[27], were loaded from GEO database (GSE129114) and re-processed by scanpy package pipeline (version 1.9.1)[104]. Each dataset was filtered out to remove the low-quality cells: only cells with 5500 ≤ genes ≥ 10,000, 75,000 ≤ transcripts ≥ 750,000, and ≤25% ERCC reads were kept for further analysis. The filtered datasets were integrated via the CCA (canonical correlation analysis) method from Seurat[105]. PCA (principal component analysis) was performed on the scaled count matrix, and PC (principal components) responsible for the cell cycle was removed. To compute a nearest-neighbor graph, first 10 PC components and 30 neighbors were used. Further clustering and embedding were performed using UMAP (Uniform Manifold Approximation and Projection)[106] and Leiden algorithm[107] respectively. The cell type assignment to the clusters was done based on a list of well-established marker genes[27]. Visualizations were done using Scanpy and Seaborn (version 0.12.1) packages[108]. $Ankrd11$ expression distributions in CNCCs cluster were analyzed using pandas v1.5.2 for statistical summaries[109].

### Statistical analysis

All immunofluorescence as well as RNAscope and BaseScope data were obtained using at least 3 embryos per genotype from at least two independent litters. All μCT and echocardiography data were obtained using at least 3 embryos per genotype from at least three independent litters. Sample sizes ($n$) indicated in the figure legends correspond to the number of embryos analyzed. All data are presented as mean ± s.e.m.

All data (except MERFISH, described below) were subjected to normality tests using the Anderson-Darling, D'Agostino & Pearson, and Kolmogorov–Smirnov tests and were found to be normal or have insufficient sample size for these tests and were thus considered normal, except Fig. S1c. For two group comparisons, two-tailed unpaired student's t-test or two-tailed multiple t-tests with Holm-Sidak multiple comparisons were used to assess statistical significance between means, where a $p$-value < 0.05 was considered significant. For three or more group comparisons two-way ANOVA was followed by Sidak's multiple comparisons test. For results that did not pass normality tests, the Mann-Whitney test was used. In all cases, Prism (version 8.0.2) was used. Sample size and statistical information are stated in the corresponding figure legends. In the figures, asterisks denote statistical significance marked by *$p$ < 0.05; **$p$ < 0.01; ***$p$ < 0.001. The sample size was not predetermined through statistical tests. In Fig. 2f one $Ankrd11^{ctrl}$ data point and in Fig. 2g one $Ankrd11^{ncko}$ data point were excluded using the ROUT method with $Q = 1\%$ to identify outliers. As $Ankrd11^{ncko}$ samples were easily identifiable by gross heart morphological abnormalities, blinded analysis was not possible. Due to the full penetrance of the phenotype regardless of the sex of the embryo, sex-based analysis was not performed.

### MERFISH data analysis

Data generated from MERSCOPE runs was first ingested in Python version 3.11.6 using the squidpy 1.3.1 library[110] User input (via the matplotlib library[111]) was then utilized to achieve two tasks: first a rotation was applied to spatial coordinates of all presented samples such that the sample "head" was pointing upwards in plots; secondly, a polygonal bounding box was defined around the OFT samples, and all cells were classified as within or outside this bounding box. All previously described data was then saved to disk. The ingest.py script contains all relevant code for the above-described steps[112].

Data saved to disk was then read using the Seurat 5.0.1 library in R version 4.3.3[113]. Only cells within the previously defined OFT bounding box were considered for further analysis. Raw RNA counts were then normalized and scaled, using the Seurat NormalizeData and ScaleData functions respectively. After the computation of principal components using the RunPCA function, a nearest-neighbor graph was constructed using the first 30 principal components via the FindNeighbors function. Cells were then blindly clustered using the Leiden algorithm[107] via the FindClusters function. Finally, 2-dimensional UMAP-space

embeddings were computed via the RunUMAP function, using the first 30 principal components.

For further analysis, it was of interest to identify cardiac neural crest cells. Using canonical CNCC markers *Penk* and *Sox9* and mesenchymal markers *Pdgfra* and *Postn*[4,47,53], clusters numbered 2, 3, and 7 were determined to be CNCCs. Differentially expressed genes between the *Ankrd11*[nchet] and *Ankrd11*[ncko] genotypes were then computed using the FindMarkers function separately for each of these three clusters using the Wilcoxon rank-sum test. The analysis.r script contains all relevant code for the above-described steps[112].

*Sema3c* images were generated using the MERSCOPE Vizualizer Version: 2.3.3330.0.

### Reporting summary

Further information on research design is available in the Nature Portfolio Reporting Summary linked to this article.

### Data availability

The scRNAseq data from ref. 27 used in this study are available in the GEO database under accession code GSE129114. The MERFISH post segmentation cell/gene expression count matrices and associated per-cell metadata generated in this study have been deposited in the GEO database under accession code GSE258835. For further data exploration, please refer to our open-science website: [https://singlocell.openscience.mcgill.ca/NeuralCrestAnkrd11-KO] (AH59.5 = ncHET 1; AH59.6 = ncHET 2; AIO4.2 = ncHET 3; AH50.1 = ncKO 1; AH59.2 = ncKO 2; AI13.3 = ncKO 3). Source data are provided with this paper.

### Code availability

The MERFISH analysis source code as well as processed and metadata is available in GitHub [https://doi.org/10.5281/zenodo.10998925].

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

## Acknowledgements

Y.K. was supported by University of Alberta Motyl Scholarship in cardiac sciences, Women and Children's Health Research Institute (WCHRI) Graduate studentship and CIHR CGS-D; N.L.D. was supported by Brad Mates E Drive Studentship in Parkinson's Disease and Movement Disorders and NSERC PGS-D; K.G. by NSERC CGS-M and WCHRI graduate studentship; AESW by CIHR CGS-D and IP by Marie Skłodowska-Curie Innovative Training Networks (ITN) Program Neu-Crest (grant agreement No. 860635). A.V. was also supported by the Canada Research Chair in Neural Stem Cell Biology and Sloan Fellowship in Neuroscience awards. This work was supported by private donations from families affected by KBG syndrome (A.V.), Gilbert Winter K. Fund from the University of Alberta Hospital Foundation operating grant (A.V.), KBG Foundation seed funding (A.V.), Stollery Children's Hospital Foundation and the Alberta Women's Health Foundation through the Women and Children's Health Research Institute grant 3925 (A.V.), European Research Council Synergy grant KILL-OR-DIFFERENTIATE 856529 (I.A.), Knut och Alice Wallenbergs Stiftelse consortium grant (IA), Vetenskapsrådet (grant 2023-02161, I.A.), Cancerfonden Project Grant (20240101, I.A.), Paradifference foundation grant (I.A.), Bertil Hållstens Forskningsstiftelse (I.A.), Göran Gustafssons Stiftelse för Naturvetenskaplig och Medicinsk Forskning (I.A.) and Austrian Science Fund Project Grant (P34136, I.A.) and Consortium Grant (SFB-F78, I.A.). The authors thank Sarah Hughes, Heather McDermid, Jonathan Epp, Dylan Terstege, Karim Fouad, Pamela Raposo, Daniela Roth, Tim Footz, Sana Bibi, and Meghan Riddell for technical assistance and helpful discussions, Alexis Allot for hosting datasets on an open science platform and Sarah Hughes for access to the confocal microscope. Finally, we thank the Advanced Cell

Exploration Core led by Mike Wong at the University of Alberta for their help with MERFISH assays.

## Author contributions

Conceptualization, Y.K. and A.V.; Methodology, Y.K., E.A., J.M., A.E.S.W., R.N., M.A., I.P., N.D., A.G., K.G., J.U., I.A., D.G., S.B., J.S., A.V.; Formal analysis, Y.K., E.A., A.E.S.W., R.N., I.P., K.G.; Resources, A.V.; Writing—original draft, YK; Writing—review & editing, Y.K. and A.V.; Supervision, A.V., J.U., I.A., D.G., S.B., J.S.; Funding acquisition, A.V.

## Competing interests

The authors declare the following competing interests: J.M. is an employee of Vizgen. The remaining authors declare no competing interests.
