## [Peer Review File · Nature Communications]

The chromatin regulator Ankrd11 controls cardiac neural crest cell-mediated outflow tract remodeling and heart functionREVIEWER COMMENTS

Reviewer #1 (Remarks to the Author):

In this manuscript, Kibalnyk and colleagues study *Ankrd11* in cardiac neural crest cells. They knock out the gene using the *Wnt1Cre2* allele and characterize the cardiac defects caused by loss of *Ankrd11*. Their data show that loss of *Ankrd11* in neural crest cells results in outflow tract defects (though the authors may be more precise whether the types of conotruncal defects observed in their mice are also observed in humans). Their analysis nominates at least 3 pathways which are dysregulated upon *Ankrd11* deletion. Overall, the manuscript is well-written and the figures easy to follow. The immunohistochemistry is of high quality and quantification of various defects is provided. My major concern with the manuscript relate to the conceptual advance provided by the manuscript. For example, it is that it is unclear if any of the pathways shown to be dysregulated in the mutant embryos are causative of phenotype. I think experiments trying to address that point would substantially improve the impact of the manuscript. Major points for the authors to consider:

1. The authors should provide evidence that *Ankrd11* transcript, or ideally protein, is absent in neural crest cells.
2. The authors implicate three candidate pathways, *Sema3C*, BMP and mTOR - on the basis of differences in expression levels observed in immunohistochemistry. The strength of these findings would be bolstered with orthogonal evidence of pathways involvement. For example - immunohistochemistry of target genes and/or qPCR/immunoblotting for relevant pathway effectors from dissected tissue.
3. The conceptual impact of the manuscript would be significantly improved if the authors could nominate a causal pathway in development in (some) aspect of the phenotypes observed. Could the authors turn to an explant system and determine whether restoration of BMP or mTOR levels prevents aspects of the phenotype? For example, given the OFT cushion defects, are valve defects observed (as observed in patients) and if so, is there a loss of EMT? Is this mediated by BMP and/or mTOR?
4. The *Sema3C* RNA-scope is difficult to visualize. The authors may consider changing the pseudocolor of the *Sema3C* probe and DAPI to better visualize the *Sema3C* signal.

Reviewer #2 (Remarks to the Author):

In this paper, Kibalnyk and colleagues characterize the role of the chromatin regulator gene *Ankrd11* in cardiovascular development. By using a combination of μ CT and mouse genetics, the authors show that homozygous loss of *Ankrd11* results in persistent truncus arteriosus, amongst other cardiac abnormalities, with 100% penetrance. They hypothesize that this phenotype is not a result of aberrant proliferation or apoptosis of cardiac neural crest cells, but that of delayed organization in the outflow tract. Finally, through immunostainings for two effector proteins, pSmad and pS6, the authors conclude that BMP and mTOR signaling pathways play a role in proper outflow tract septation.

While the title and introduction promise to tackle the very important question of how chromatin regulators affect cardiac neural crest development, the manuscript falls short of addressing this question. Instead, the authors simply describe the phenotypes they observed in a heterozygous and homozygous mutant for *Ankrd11*. This mouse model appears to have been previously described by this lab in a recent 2021 paper, where they looked at craniofacial defects in *Ankrd11* mutant background. The conclusions drawn by the paper, in the absence of any mechanistic insights on how *Ankrd11* interacts with BMP and mTOR signaling, are a mere validation of previous studies that have shown these signaling pathways to be important for cardiovascular development. Even the analysis in

figure 5N was previously described by Darrigrand and colleagues. It is unclear to this reviewer as to how this paper advances the field and fits within the scope of Nature Communications. I encourage the authors to work out the mechanism by which Ankrd11 is recruiting BMP and/or mTOR signaling in regulating OFT septation.

More specific comments and issues are listed below:

1. Lines 122-124: The authors use Fig.S1C-E as evidence for Ankrd11 expression in the cardiac neural crest. However, it seems that only a subset of cells have any RNAscope signal. Using the authors' approach of quantifying RNAscope signal in sections, I wonder what percentage of cardiac crest cells actually express Ankrd11? The scRNA-seq from the Adameyko lab would suggest that a greater proportion of cells express this gene, at least at earlier stages. Is Ankrd11 lost from the cardiac crest? If so, how does that explain the phenotypes described in the paper? Please quantify the signal and report what proportion of cells express Ankrd11.
2. Lines 142-144: What is the number/proportion of SMA+/YFP+ cells in WT and Ankrd11ncko mutant embryos? Please quantify and report the difference, or lack thereof, in the paper.
3. Lines 191-192: The conclusion authors draw here is confusing. Why would a greater proportion of the outflow tract be colonized by cardiac neural crest cells (supp figure 3d) if the overall number of cardiac neural crest cells is reduced (figures 3-4)? On a similar note, how would the observation of unperturbed distance of CNCC migration into the outflow tract be explained by "...a modest defect in migration rather than abnormal proliferation or apoptosis..."?
4. What structural features were used to define proximal, medial, and distal regions of the outflow tract? Given the marginally significant differences reported in figures 3 and 4, this reviewer is concerned that the results may have a confounding variable.
5. Lines 539-540: On that note, how were the YFP+ cells quantified? More information in the materials and methods section is warranted. Given that the mesenchymal cells in the heart tend to condense, segmenting cytoplasmic signal can result in quantification errors.
6. If the cardiac neural crest occupation of the outflow tract seemed to recover in the mutant background, could the septation process also be delayed? How late was the phenotype characterized in sections?
7. It is unclear why the authors chose to only quantify their phenotypes at the medial level of the outflow tract in figure 7. Also, the pSmad fluorescence intensity should be normalized to the number of YFP+ cells within the ROI and reported as a proportion similar to panel 7N to rule out potential effects due to reduced cell numbers.
8. This reviewer would like to see the raw data that was used to perform statistical analysis to ensure that conditions for using a parametric test on a small sample size were met. The data plotted in the figures suggest unequal variance between samples, which doesn't seem to have been considered in the authors' choice of statistical test.

Reviewer #3 (Remarks to the Author):

In this study the authors claim that Ankrd11 is a critical regulator of heart development. Ankrd11 is a chromatin regulator related to KBG syndrome which clinical manifestation includes, macrodontia, craniofacial findings, short stature, skeletal anomalies, global developmental delay, seizures, and intellectual disability. Heart anomalies were initially described in a minor proportion (Sirmaci, A., et al.

Am. J. Hum. Genet. 89: 289-294, 2011), but now is found in ~44% of KBG patients (Digilio MC, et al.. Am J Med Genet A. 2022 Apr;188(4):1149-1159).

- What are the noteworthy results?

The authors showed that ablation of *Ankrd11* in the murine neural crest is related to aberrant heart development and function. This is the main result. In addition, the characterization of abnormal signaling pathways pinpoint the molecular mechanism that may be involved.

- Will the work be of significance to the field and related fields? How does it compare to the established literature? If the work is not original, please provide relevant references.

The work is original. The cardiac defects are more severe than the ones observed in KBG syndrome, but that difference may be related to gene dosage.

- Does the work support the conclusions and claims, or is additional evidence needed

1. One concern is that the ablation of *Ankrd11* is very detrimental by itself. *Ankrd11* KO animals don't survive. So, maybe the ablation of *Ankrd11* is killing the cardiac neural crest cells. It seems not to be the case since YFP markers are present in the ncko embryos, but YFP expression is a reporter for *wnt1* expression, and maybe ectopic expression of *wnt1* is marking other cells.

2. It is known that ablation of neural crest cells resulted in persistent truncus arteriosus, mis-patterning of the great vessels, outflow malalignments, and hypoplasia or aplasia of the pharyngeal glands. If *Ankrd11* ablation in the neural crest cells is not killing the cells but is affecting exclusively the cardiovascular development then hypoplasia of thymus, thyroid, and parathyroid glands, should not be present. How are the thymus, thyroid, and parathyroid glands in the ncko animals? If they are normal, this will reinforce that the effect is on cardiac development specifically. I understand that craniofacial abnormalities are also seen in these mice, but those are also seen in KBG syndrome, hence expected.

3. Something that maybe problematic is that when you look into the information on B6.Cg-E2f1Tg(*Wnt1-Cre*)2Sor/J Strain #022501 Common Name: B6 *Wnt1-Cre2* <https://www.informatics.jax.org/allele/MGI:5485027?recomRibbon=open>): you find

cardiovascular system

pulmonary trunk hypoplasia (J:298597)

- about 5% of mutants show a hypoplastic pulmonary trunk at E15.5 and E16.5

interrupted aortic arch, type b (J:298597)

- about 30% of mutants exhibit interrupted aortic arch, type b at E15.5 and E16.5

abnormal conotruncal ridge morphology (J:298597)

- the number of neural crest cells in the proximal outflow tract cushions is reduced at E11.5

double outlet right ventricle (J:298597)

- all mutants exhibit the double outlet right ventricle at E15.5 and E16.5

ventricular septal defect (J:298597)

- all mutants exhibit a ventricular septal defect at E15.5 and E16.5

If <https://www.jax.org/strain/022501> is not the strain used, please clarify.

4. *Ankrd11* ncko embryos (*Ankrd11*^{fl}/WT; *Wnt1*Cre2) showed normal OFT septation, how many embryos were analyzed? There are some haploinsufficiency *Ankrd11* related to KBG syndrome presenting heart abnormalities (Digilio MC, et al.. Am J Med Genet A. 2022 Apr;188(4):1149-1159). It is strange that no defect is seen in heterozygous animals.

Minor issues:

1. Embryonic hearts at E18.5 were analyzed for anatomical anomalies: all *Ankrd11* ncko embryos exhibited a persistent truncus arteriosus (PTA). How many?

2. Supplemental Figure 3: not sure that the image for the wt and the *Ankrd11* ncko embryos are in the same plane, hence comparable.

3. Figure 3 A and D seem to show a different area for the distal part (white squares).

- Are there any flaws in the data analysis, interpretation and conclusions? Do these prohibit publication or require revision?

If the authors can explain the points raised, I think it is a good work. Main concern is if the phenotype

observed (that is very well documented) is related to Ankrd11 ablation.

- Is the methodology sound? Does the work meet the expected standards in your field?

Yes

- Is there enough detail provided in the methods for the work to be reproduced?

Yes

REVIEWER COMMENTS

Reviewer #1 (Remarks to the Author):

In this manuscript, Kibalyk and colleagues study *Ankrd11* in cardiac neural crest cells. They knock out the gene using the *Wnt1Cre2* allele and characterize the cardiac defects caused by loss of *Ankrd11*. Their data show that loss of *Ankrd11* in neural crest cells results in outflow tract defects (though the authors may be more precise whether the types of conotruncal defects observed in their mice are also observed in humans). Their analysis nominates at least 3 pathways which are dysregulated upon *Ankrd11* deletion. Overall, the manuscript is well-written and the figures easy to follow. The immunohistochemistry is of high quality and quantification of various defects is provided.

We thank the reviewer for providing their constructive feedback and for highlighting the potential of our results. Below we have addressed all raised concerns.

My major concern with the manuscript relate to the conceptual advance provided by the manuscript. For example, it is that it is unclear if any of the pathways shown to be dysregulated in the mutant embryos are causative of phenotype. I think experiments trying to address that point would substantially improve the impact of the manuscript. Major points for the authors to consider:

1. The authors should provide evidence that *Ankrd11* transcript, or ideally protein, is absent in neural crest cells.

We thank the reviewer for this suggestion. Unfortunately, there are no good commercial antibodies raised against *Ankrd11* that we have been able to validate. Some custom antibodies existed, but they are no longer available. Instead, we have addressed this concern by performing BaseScope, a single-molecule RNA fluorescent *in situ* hybridization technique and showed that ~80% of the control (*Ankrd11^{nchet}*) outflow tract (OFT) mesenchyme cells expressed *Ankrd11* mRNA. In contrast, ~20% of *Ankrd11^{ncko}* OFT mesenchyme cells expressed *Ankrd11* mRNA (new Fig. 1i-j', l). Please note that technical limitations prevented us from counterstaining against YFP to identify neural crest cells, however, the OFT mesenchyme is predominantly composed of YFP+ cells¹. The residual *Ankrd11* signal in *Ankrd11^{ncko}* OFT mesenchyme could be due to the presence of second heart field (SHF), non-neural crest derived cells in the OFT mesenchyme¹, or incomplete recombination. Notably, a Basescope assay with a negative control probe did not yield a signal (new Fig. 1k). Furthermore, in response to reviewer #2, we further stratified *Ankrd11*-expressing cells in the control OFT mesenchyme based on the *Ankrd11* expression level (new Fig. 1m). Our results demonstrate low, mid, and high *Ankrd11*-expressing cells (new Fig. 1m). Overall, our BaseScope results are in line with expanded re-analysis of published neural crest single-cell RNA sequencing dataset that also demonstrates low, mid and high *Ankrd11*-expressing CNCCs (former Fig. S1, now updated Fig. 1a-e).

We have reflected new data in Fig. 1 and associated text in “Materials and Methods” on pages 16-17, 19-20, and “Results” on page 5

2. The authors implicate three candidate pathways, *Sema3C*, *BMP* and *mTOR* - on the basis of differences in expression levels observed in immunohistochemistry. The strength of these findings would be bolstered with orthogonal evidence of pathways involvement. For example - immunohistochemistry of target genes and/or qPCR/immunoblotting for relevant pathway effectors from dissected tissue.

Thank you for the insightful suggestion. The OFT displays distinct gene expression and signalling pathway activation in distal, medial and proximal areas ². Therefore, we could not utilize qPCR or immunoblotting techniques from the entire OFT tissue as this does not permit a spatial analysis. We have therefore opted to use some additional immunohistochemistry analysis at E11.5 and E12.5 and have expanded our E11.5 OFT analysis to the single cell spatial transcriptomics technique MERFISH (Multiplexed Error-Robust Fluorescence in situ Hybridization) with a custom 140 target gene panel. This custom panel included genes that are known to be important for OFT development and/or that were previously shown to be expressed in the OFT during the septation process in a scRNAseq dataset ²⁻¹². These include factors of important signaling pathways, extracellular matrix components, chemokines, transcription and chromatin remodeling factors, and cell type markers (new Table S1).

With regard to the BMP pathway, MERFISH analysis identified reduced expression of *Bmp4*, *Bmpr2*, and *Acvr1* in E11.5 *Ankrd11^{ncko}* CNCCs (new Fig. 8a-c). Furthermore, results in ² suggest that BMP signaling at least in part controls *Sema3c* expression. We found downregulated *Sema3c* in E11.5 *Ankrd11^{ncko}* CNCCs using MERFISH, which we corroborated with RNAscope (new Fig. 8a-c & S6), indicating a potential downstream effect of impaired BMP signaling. Finally, in response to reviewer #2, we have re-analyzed E11.5 IHC images of pSmad1/5/8, the intracellular effectors of Smad-dependent BMP signaling ^{2,13,14}. New analysis shows reduced proportion of pSmad1/5/8+ CNCCs in E11.5 *Ankrd11^{ncko}* medial OFT compared to *Ankrd11^{nchet}* (former Fig. 7, now updated Fig. 4). Altogether, this supports and expands our original conclusion that the BMP pathway is deregulated in *Ankrd11^{ncko}* CNCCs.

pS6 remains the gold standard readout of mTOR signaling ^{15,16}, which is what we used in our original submission. Notably, the involvement of the mTOR pathway in OFT development is a recent discovery ¹². As such, there are no known direct mTOR targets for cardiac neural crest cells to date. However, Nie 2021 identified that mTOR deletion causes dysregulation of actin dynamics and Smad1/5/8 phosphorylation ¹², which were both affected in *Ankrd11^{ncko}* OFTs (Former Figs. 5, 7; now updated Figs. 3-5), suggesting a potential crosstalk between mTOR and BMP signaling.

The MERFISH assay revealed several differentially expressed genes (DEGs) in *Ankrd11^{ncko}* CNCCs for various signaling pathways known to be important for OFT development, including Wnt (*Fzd4*, *Vangl2*, *Fzd1*), BMP (*Id2*, *Bmp4*, *Smad7*, *Bmpr2*, *Acvr1*), TGF- β (*Tgfb2*, *Smad7*), Notch (*Heyl*, *Jag1*, *Notch1*), Hippo (*Yap1*), JNK (*Jun*), Retinoic acid (*Rxra*, *Rarg*), FGF (*Fgfr2*), EGFR (*ErbB3*), and JAK-STAT (*Stat3*) ²⁻¹² (new Fig. 8a-c and Table S2). Furthermore, it identified downregulation in neuropilin-1 (*Nrp1*), a known receptor for *Sema3c* ⁴, which may amplify the effects of the dysregulation (please see above). Other chemokines and growth factors included ephrin A5 (*Efna5*) and transforming growth factor beta 2 (*Tgfb2*) ^{17,18} (new Fig. 8a-c and Table S2). DEGs also included other transcription and chromatin remodeling factors important for OFT development such as *Egr1*, *Hoxa3*, *Ets1*, *Gata4*, *Sox11*, *Foxc1* and *Chd7* ^{10,19,20}, and cytoskeletal or extracellular matrix factors, such as *Cdh11*, *Mmp14*, *Adamts1*, and *Adam19* ²¹ (new Fig. 8a-c and Table S2). These results suggest that *Ankrd11* ablation affects multiple signaling pathways, transcription factors and chromatin remodelers important for CNCC function and OFT morphogenesis.

We expanded our IHC analysis to additional signalling pathways and molecules to corroborate some of the MERFISH results. We focused on phosphorylated Smad2/3, the intracellular effectors of Smad-dependent TGF- β signaling ²², and Cellular retinoic acid binding protein 2 (Crabp2), a protein that transports retinoic acid from the cytoplasm to the nucleus, which facilitates retinoic acid signaling ²³. We observed asymmetrical pSmad2/3 and Crabp2 signal in control medial parietal (top-right) and septal (bottom-left) OFT cushions at E11.5 (new Figs. 6 and S4). In contrast, E11.5 *Ankrd11^{ncko}* medial

OFT parietal and septal cushions had similar levels of pSmad2/3 or Crabp2 (new Figs. 6 and S4). While nothing is known about the mechanism behind the pSmad2/3 and Crabp2 asymmetry within CNCCs or its downstream effects on AP septum formation, there is evidence for CNCC asymmetry. Gandhi et al²⁴ observed that ablation of the right cardiac neural folds in chick, which removes CNCCs from the right side of the embryo, produced a more severe septation impairment compared to ablation of the left cardiac neural folds.

Please note that in response to reviewer #2, we have expanded pS6, pSmad1/5/8, pSmad2/3 and actin dynamics analysis to E12.5 (updated Figs. 3-5 and S3, new Fig. 6). Overall, our results show comparable levels of pS6, pSmad1/5/8 and pSmad2/3 in E12.5 Ankrd11^{nchet} and Ankrd11^{ncko} medial OFT cushions (distal cushions were analyzed as well, but only in Ankrd11^{ncko} samples), suggesting a delay in mTOR, BMP and TGF- β signalling pathways and CNCC condensation.

Overall, we propose that several dysregulated pathways contribute to Ankrd11^{ncko} CNCC delay in condensation and convergence at the midline to form the AP septum, including BMP, mTOR, and TGF- β .

We have reflected new data in Figs. 4-8, S3-4 and associated text in “Materials and Methods” on pages 16-22, “Results” on pages 8-11 and “Discussion” on pages 12-14.

3. The conceptual impact of the manuscript would be significantly improved if the authors could nominate a causal pathway in development in (some) aspect of the phenotypes observed. Could the authors turn to an explant system and determine whether restoration of BMP or mTOR levels prevents aspects of the phenotype? For example, given the OFT cushion defects, are valve defects observed (as observed in patients) and if so, is there a loss of EMT? Is this mediated by BMP and/or mTOR?

Thank you for this suggestion. Neural crest cells participate in a strict spatiotemporal control of tissue (OFT) morphogenesis. It has proven difficult to recapitulate our main findings (failure of OFT septation) in an explant culture. We have attempted a neural tube explant to measure Ankrd11^{ncko} neural crest migration, however preliminary results did not show a migration defect (data not shown). This is corroborated by our *in vivo* findings, where we found small and region-specific decreases in CNCC number in the OFT. Furthermore, the explant system cannot faithfully recapitulate the morphogenesis of AP septation, which is the main affected phenotype in our manuscript.

Moreover, the MERFISH assay shows a broad set of genes and signalling pathways to be dysregulated in Ankrd11^{ncko} CNCCs (please see above and new Figs. 7-8, Tables S1-2). It is thus not feasible to name a nominal pathway as our data suggest ablation of Ankrd11 leads to perturbations in various signalling pathways, extending beyond the originally nominated BMP and mTOR signalling pathways in our original submission. Finally, due to signalling pathway cross-talk, it is difficult to choose a single nominal pathway and rescue strategy.

Nevertheless, we have attempted pharmacological rescues *in vivo*, using a BMP agonist Sb4²⁵ and an mTOR agonist L-leucine²⁶ administration into pregnant dams. Due to previous research showing that Ankrd11 represses the retinoic acid receptor RAR α ²⁷, we also used retinoic acid signaling inhibitor BMS493²⁸. None of the experiments resulted in a rescue in Ankrd11^{ncko} embryos (data not shown) and overall proved to be extremely difficult to interpret. This is at least in part due to the pan-effects of these drugs on all tissues, not just CNCCs. As CNCCs and surrounding tissues participate in cross-talk

during OFT morphogenesis, the drugs affect all tissues with active signalling pathway of choice. Moreover, BMP, mTOR and/or RA signalling is vital for proper embryogenesis. Thus, interpretation of results even in control embryos would be very challenging.

We have therefore re-focussed our efforts on MERFISH and additional IHC analysis. To our knowledge, our report is the first one to apply spatial single cell transcriptomics to OFT septation. Our dataset will be useful for the community-at-large and will provide a rich resource for data mining in control and *Ankrd11*-deficient embryos (please note that the datasets will be deposited into a public repository and an open visualization website prior to publication; meanwhile, we have uploaded processed and meta-data alongside source code via GitHub). Furthermore, we provide evidence for asymmetric signaling in control OFT cushions involving the TGF- β and retinoic acid pathways, which to our knowledge has not been described previously.

Overall, our new and expanded results support and extend our work, suggesting that *Ankrd11* controls a complex network of genes required for OFT development.

Finally, we have also analyzed valve development and observed dysplastic valves in 3 of 5 *Ankrd11*^{neko} hearts (new Fig. S1d-i). Since these valves appear hypertrophic, this does not suggest a loss of EMT, but rather a failure of the CNCCs to remodel the valves correctly²⁹, echoing their failure to remodel the AP septum. Based on our OFT signaling pathway results using IHC and MERFISH (updated and/or new Figs. 4-8), we suspect that this is caused by impairment of multiple pathways.

We have reflected new data in Figs. S1, S3-4, 6-8, Tables S1-S2, and associated text in “Materials and Methods” on pages 16-22, “Results” on pages 6, 8-11 and “Discussion” on pages 11-14.

4. The *Sema3C* RNA-scope is difficult to visualize. The authors may consider changing the pseudocolor of the *Sema3C* probe and DAPI to better visualize the *Sema3C* signal.

Thank you for this comment. This has been addressed in new Fig. S6 (former Fig. S4). Moreover, we have added representative *Sema3c* MERFISH images to this figure, which corroborate our RNA scope findings (MERFISH images in new Fig. S6A, MERFISH results in new Fig. 8).

Reviewer #2 (Remarks to the Author):

In this paper, Kibalnyk and colleagues characterize the role of the chromatin regulator gene *Ankrd11* in cardiovascular development. By using a combination of μ CT and mouse genetics, the authors show that homozygous loss of *Ankrd11* results in persistent truncus arteriosus, amongst other cardiac abnormalities, with 100% penetrance. They hypothesize that this phenotype is not a result of aberrant proliferation or apoptosis of cardiac neural crest cells, but that of delayed organization in the outflow tract. Finally, through immunostainings for two effector proteins, pSmad and pS6, the authors conclude that BMP and mTOR signaling pathways play a role in proper outflow tract septation.

While the title and introduction promise to tackle the very important question of how chromatin regulators affect cardiac neural crest development, the manuscript falls short of addressing this question. Instead, the authors simply describe the phenotypes they observed in a heterozygous and

homozygous mutant for *Ankrd11*. This mouse model appears to have been previously described by this lab in a recent 2021 paper, where they looked at craniofacial defects in *Ankrd11* mutant background. The conclusions drawn by the paper, in the absence of any mechanistic insights on how *Ankrd11* interacts with BMP and mTOR signaling, are a mere validation of previous studies that have shown these signaling pathways to be important for cardiovascular development. Even the analysis in figure 5N was previously described by Darrigrand and colleagues. It is unclear to this reviewer as to how this paper advances the field and fits within the scope of Nature Communications. I encourage the authors to work out the mechanism by which *Ankrd11* is recruiting BMP and/or mTOR signaling in regulating OFT septation.

Thank you for your constructive feedback and recommendation. Reviewer #1 has also recommended to “nominate a causal pathway in development in (some) aspect of the phenotypes observed”. Our combined response is below:

Neural crest cells participate in a strict spatiotemporal control of tissue (OFT) morphogenesis. We were unable to use an *in vitro* system to interrogate the mechanism of *Ankrd11* in OFT morphogenesis because the explant system cannot faithfully recapitulate the morphogenesis of AP septation, which is the main affected phenotype in our manuscript. For example, we have attempted a neural tube explant to measure *Ankrd11*^{ncko} neural crest migration, however preliminary results did not show a migration defect (data not shown). This is corroborated by our *in vivo* findings, where we found small and region-specific decreases in CNCC number in the OFT. We were thus unable to use explant or other *in vitro* systems to study the mechanism.

However, we performed two new major sets of *in vivo* experiments to address the mechanism.

First, we performed single cell spatial transcriptomics MERFISH (Multiplexed Error-Robust Fluorescence in situ Hybridization) with a custom 140 target gene panel. This custom panel included genes that are known to be important for OFT development and/or that were previously shown to be expressed in the OFT during the septation process in a scRNAseq dataset²⁻¹². These include factors of important signaling pathways, extracellular matrix components, chemokines, transcription and chromatin remodeling factors, and cell type markers (new Table S1). This technique was chosen due to spatially controlled gene expression in distal, medial and proximal OFT areas². Therefore, we could not utilize scRNA-seq, qPCR or immunoblotting techniques from entire OFT tissue as they do not permit a spatial analysis.

The MERFISH assay shows a broad set of genes and signalling pathways to be dysregulated in *Ankrd11*^{ncko} CNCCs (new Figs. 7-8, Tables S1-2). In summary, MERFISH assay revealed several differentially expressed genes (DEGs) in *Ankrd11*^{ncko} CNCCs for several signaling pathways known to be important for OFT development, including Wnt (*Fzd4*, *Vangl2*, *Fzd1*), BMP (*Id2*, *Bmp4*, *Smad7*, *Bmpr2*, *Acvr1*), TGF- β (*Tgfb2*, *Smad7*), Notch (*Heyl*, *Jag1*, *Notch1*), Hippo (*Yap1*), JNK (*Jun*), Retinoic acid (*Rxra*, *Rarg*), FGF (*Fgfr2*), EGFR (*ErbB3*), and JAK-STAT (*Stat3*)²⁻¹² (new Fig. 8a-c and Table S2). We found DEGs in several growth factors and chemokines. In cluster 2, one of the most downregulated DEGs in *Ankrd11*^{ncko} cells was *Sema3c*, which we have corroborated with single molecule FISH (RNA scope) (new Fig. S6). The class 3 Semaphorin C (Sema3C) plays a major role in CNCC migration into the OFT, their condensation in the OFT cushions, and their convergence to create the AP septum, at least in part due to its role as a chemoattractant and aggregation factor^{2,30-33}. Furthermore, the assay identified downregulation in neuropilin-1 (*Nrp1*), a known receptor for Sema3C⁴, which may amplify the effects of the dysregulation. Other factors included ephrin A5 (*Efna5*), which is known to guide neural crest migration, and transforming growth factor beta 2 (*Tgfb2*), which is

important for OFT morphogenesis^{17,18} (new Fig. 8a-c and Table S2). DEGs also included other transcription and chromatin remodeling factors important for OFT development such as *Egr1*, *Hoxa3*, *Ets1*, *Gata4*, *Sox11*, *Foxc1* and *Chd7*^{10,19,20}, and cytoskeletal or extracellular matrix factors, such as *Cdh11*, *Mmp14*, *Adamts1*, and *Adam19*²¹ (new Fig. 8a-c and Table S2). These results suggest that Ankrd11 ablation affects multiple signaling pathways, transcription factors and chromatin remodelers important for CNCC function and OFT morphogenesis, further supporting our expanded IHC results in updated or new Figs. 4-6.

Thus, it is unlikely that Ankrd11 controls OFT morphogenesis via BMP and/or mTOR signalling pathways only. It is more likely that Ankrd11 knockout leads to dysregulation of many signalling pathways and their cross-talk.

In addition, and in response to this reviewer's additional concerns below, we have extended our IHC analysis to distal OFT regions at E11.5 and included medial and distal OFT analysis at E12.5. This analysis showed spatiotemporal differential dysregulation of mTOR (pS6), BMP (pSmad1/5/8), TGF- β (pSmad2/3) and RA (Crabp2) signaling pathways (updated and new Figs. 4-6, S3-4). Thus, further mechanistic interrogation of how Ankrd11 recruits BMP, mTOR or other dysregulated signaling during OFT septation will require novel spatial tools as using traditional entire OFT primary cells or explants will not yield spatially resolved specimens for appropriate interpretation of the results. Such further mechanistic work lies outside of the scope of this manuscript.

Second, we have attempted pharmacological rescues *in vivo*, using a BMP agonist Sb4²⁵ and an mTOR agonist L-leucine²⁶. Due to previous research showing that Ankrd11 represses the retinoic acid receptor RAR α ²⁷, we also used retinoic acid signaling inhibitor BMS493²⁸. None of the experiments resulted in a rescue in Ankrd11^{ncko} embryos (data not shown) and overall proved to be extremely difficult to interpret. This is at least in part due to the pan-effects of these drugs on all tissues, not just CNCCs. As CNCCs and surrounding tissues participate in cross-talk during OFT morphogenesis, the drugs affect all tissues. Moreover, BMP, mTOR and/or RA signalling is vital for proper embryogenesis. Thus, interpretation of results even in control embryos would be very challenging. Finally, our MERFISH results show a broad set of genes and signalling pathways to be dysregulated in Ankrd11^{ncko} CNCCs (please see above and new Figs. 7-8, Tables S1-2). It is thus not feasible to name a nominal pathway as our data suggest ablation of Ankrd11 leads to perturbations in various signalling pathways, extending beyond the originally nominated BMP and mTOR signalling pathways in our original submission.

We have therefore re-focussed our efforts on MERFISH and additional IHC analysis. To our knowledge, our report is the first one to apply spatial single cell transcriptomics to OFT septation. Our dataset will be useful for the community-at-large and will provide a rich resource for data mining in control and Ankrd11-deficient embryos (please note that the datasets will be deposited into a public repository and an open visualization website prior to publication; meanwhile, we have uploaded processed and meta-data alongside source code via GitHub). Furthermore, we provide evidence for asymmetric signaling in control OFT cushions involving the TGF- β and retinoic acid pathways, which to our knowledge has not been described previously. Finally, our new results support and expand our original conclusions, and suggest that Ankrd11 controls a complex network of genes required for OFT development. These new datasets have shown known, and more importantly, novel signalling pathways that Ankrd11 controls. In our future work, we will study an in-depth mechanism by which Ankrd11 controls these signalling pathways and gene expression. However, this is outside of the scope of the current manuscript.

We have reflected new data in Figs. 4-8, S3-S4, Tables S1-S2, and associated text in “Materials and Methods” on pages 16-22, “Results” on pages 8-11 and “Discussion” on pages 12-14.

More specific comments and issues are listed below:

1. Lines 122-124: The authors use Fig.S1C-E as evidence for *Ankrd11* expression in the cardiac neural crest. However, it seems that only a subset of cells have any RNAscope signal. Using the authors’ approach of quantifying RNAscope signal in sections, I wonder what percentage of cardiac crest cells actually express *Ankrd11*? The scRNA-seq from the Adameyko lab would suggest that a greater proportion of cells express this gene, at least at earlier stages. Is *Ankrd11* lost from the cardiac crest? If so, how does that explain the phenotypes described in the paper? Please quantify the signal and report what proportion of cells express *Ankrd11*.

We thank the reviewer for this suggestion. We have addressed this concern by performing BaseScope, a single-molecule RNA fluorescent *in situ* hybridization technique and showed that ~80% of the control (*Ankrd11*^{nchet}) outflow tract (OFT) mesenchyme cells expressed *Ankrd11* mRNA (new Fig. 1i, l). Notably, a Basescope assay with a negative control probe did not yield a signal (new Fig. 1k). Please note that technical limitations prevented us from counterstaining against YFP to identify neural crest cells, however, the OFT mesenchyme is predominantly composed of YFP+ cells¹. We further stratified *Ankrd11*-expressing cells in the control OFT mesenchyme based on the *Ankrd11* expression level (new Fig. 1m). Our results demonstrate low, mid and high *Ankrd11*-expressing cells (Fig. 1m). Re-analysis of published neural crest single-cell RNA sequencing dataset also demonstrates low, mid and high *Ankrd11*-expressing CNCCs (former Fig. S1, now updated Fig. 1a-e). Notably, in comparison to ~80% of CNCCs expressing *Ankrd11* in control CNCCs, scRNAseq shows that all CNCCs express *Ankrd11*. This could be due to different methods used (BaseScope vs scRNA-seq), samples (WT in scRNA-seq vs *Ankrd11*^{nchet} in BaseScope) and/or potential inclusion of second heart field (SHF), non-neural crest derived cells in the OFT mesenchyme¹, in BaseScope analysis.

We have reflected new data in Fig. 1 and associated text in “Materials and Methods” on pages 16-17, 20, and “Results” on page 5.

2. Lines 142-144: What is the number/proportion of SMA+/YFP+ cells in WT and *Ankrd11*^{ncko} mutant embryos? Please quantify and report the difference, or lack thereof, in the paper.

Thank you for this suggestion. To address this concern, we have performed immunofluorescence analysis of cross sections of E18.5 outflow tract vessels and quantified SMA+YFP+ cells out of the total SMA+ cells within the proximal (closest to the heart) and distal regions of the vessels (new Fig. S1j-p). In the proximal region, analysis of the α SMA+ (alpha-smooth muscle actin) smooth muscle layer of the E18.5 *Ankrd11*^{ncko} truncal artery did not show a deficiency of CNCC-derived smooth muscle cells compared to the *Ankrd11*^{nchet} aortic and pulmonary arteries (new Fig. S1j-p). Although the *Ankrd11*^{ncko} distal region showed a decreased proportion of smooth muscle cells that are CNCC derived ($\% \alpha$ SMA+YFP+/ α SMA+; new Fig. S1n), this was not due to their decreased number (#YFP+ α SMA+; new Fig. S1o), but rather due to an increase in non-CNCC derived smooth muscle cells (#YFP- α SMA+; new Fig. S1p).

We have reflected new data in Fig. S1 and associated text in “Materials and Methods” on page 19, and “Results” on page 6.

3. Lines 191-192: The conclusion authors draw here is confusing. Why would a greater proportion of the outflow tract be colonized by cardiac neural crest cells (supp figure 3d) if the overall number of cardiac neural crest cells is reduced (figures 3-4)? On a similar note, how would the observation of unperturbed distance of CNCC migration into the outflow tract be explained by "...a modest defect in migration rather than abnormal proliferation or apoptosis...?"

Thank you for highlighting this and we apologize for the confusion. Former Fig. S3 showed that there are no statistically significant differences in the distance traveled by the first invading cardiac neural crest cells into the proximal outflow tract. This differs from other genetic models of cardiac neural crest deficiency, where the cells are not able to infiltrate the most proximal regions of the OFT, indicating a more severe migration defect³⁴. This indicates that the *Ankrd11^{ncko}* cardiac neural crest cells have a more subtle migration defect, where the cells were able to populate the full length of the OFT, but in smaller numbers, since proliferation or apoptosis was not affected (Fig. S2a-i; former Fig. 3). This is in line with findings from other genetic models, including Nie et al.¹², which showed that mTOR deletion led to reduced CNCC numbers in the OFT without changes in apoptosis or proliferation prior to E11.5, suggesting that the cell deficit was caused by a migration defect.

Since reviewer 3 also found concerns with these results, we removed these results from the manuscript to remove confusion, as they are not critical for our conclusions.

4. What structural features were used to define proximal, medial, and distal regions of the outflow tract? Given the marginally significant differences reported in figures 3 and 4, this reviewer is concerned that the results may have a confounding variable.

Due to the expansion of the manuscript and the similarity in the results between E10.5 and E11.25 (Figs. 3-4 in original submission), we have removed the E11.25 data from the manuscript to remove redundancy and confusion between E11.25 and E11.5 ages. E10.5 data are now presented in Fig. S2.

To address this comment, we have added a detailed description defining the OFT regions to the manuscript and expanded it to include all ages analyzed (E10.5, E11.5, E12.5).

At E10.5, the proximal OFT region has a small lumen and dispersed CNCCs, the medial region has a large lumen, dispersed CNCCs and a thickened bottom cushion, and the distal region has a large lumen, more compacted CNCCs, and a thickened bottom cushion (Fig. S2a-f). We have therefore used these features to define proximal, medial and distal regions of E10.5 OFT. To analyze anatomically matched sections of *Ankrd11^{nchet}* and *Ankrd11^{ncko}* OFTs at E11.5 and E12.5, the *Ankrd11^{ncko}* distal region was determined to be the equivalent region to the *Ankrd11^{nchet}* septated region based on the number of OFT sections it occupied, and the medial sections used for analysis were adjacent to this boundary. We used this method to avoid relying on the differing morphology between *Ankrd11^{nchet}* and *Ankrd11^{ncko}* OFTs and between embryonic ages. We have addressed this clarification in materials and methods on pages 19-20.

5. Lines 539-540: On that note, how were the YFP+ cells quantified? More information in the materials and methods section is warranted. Given that the mesenchymal cells in the heart tend to condense, segmenting cytoplasmic signal can result in quantification errors.

Cells were identified as YFP+ when YFP+ cytoplasm overlaid with a Hoechst+ nucleus. All cells were counted manually. This clarification is added on page 20.

6. If the cardiac neural crest occupation of the outflow tract seemed to recover in the mutant background, could the septation process also be delayed? How late was the phenotype characterized in sections?

Thank you for the insightful comment. We extended our analysis to E12.5 *Ankrd11^{ncko}* OFTs. Surprisingly, we found that the very distal edge of E12.5 *Ankrd11^{ncko}* OFT showed septation (new Fig. 3l-m). However, the septation failed to progress any further, as at E18.5 we observe the persistent truncus arteriosus phenotype characteristic of OFT septation failure (Fig. 2).

Furthermore, we have extended pS6 and pSmad1/5/8 and added pSmad2/3 signalling analysis to include E12.5 samples as well as distal OFT in both E11.5 and E12.5 samples. With regard to CNCC condensation, the *Ankrd11^{ncko}* CNCCs in the cardiac cushions in the unseptated distal region at E12.5 were still visibly disorganized (updated Fig. 3p-q). For BMP and mTOR signalling, while E11.5 *Ankrd11^{ncko}* showed reduced % pS6+ and % pSmad1/5/8+ medial CNCCs compared to *Ankrd11^{nchet}*, their levels were similar in E12.5 medial CNCCs (updated Figs. 4-5). For the distal region, we only analyzed *Ankrd11^{ncko}* cardiac cushions as controls have formed a septum in this area. Unseptated distal region in E12.5 *Ankrd11^{ncko}* OFT showed pS6 and pSmad1/5/8 signal that was elevated in comparison to the distal region in E11.5 *Ankrd11^{ncko}* OFT (Fig. 4k-l, 5k-l). Finally, we included new analysis of pSmad2/3, the intracellular effectors of Smad-dependent TGF- β signaling²². We found asymmetric pSmad2/3 signaling in control medial parietal (top-right) and septal (bottom-left) OFT cushions at E11.5 (new Fig. 6). In contrast, E11.5 *Ankrd11^{ncko}* medial OFT parietal and septal cushions had similar levels of pSmad2. However, by E12.5, *Ankrd11^{ncko}* medial OFT showed asymmetric pSmad2/3 staining. (new Fig. 6). As described above, for the distal region, we only analyzed *Ankrd11^{ncko}* cardiac cushions as controls have formed a septum in this area. In the distal region, at both E11.5 and E12.5, we detected comparable levels of pSmad2/3+ CNCCs between *Ankrd11^{ncko}* parietal and septal cushions (new Fig. 6). Therefore, these results show that the *Ankrd11^{ncko}* CNCCs display a delay in BMP and mTOR signalling and a delay in asymmetric activation of TGF- β signalling. Even though some signalling was recovered in E12.5 *Ankrd11^{ncko}* OFT, the *Ankrd11^{ncko}* AP septum failed to fully establish. This suggests that the *Ankrd11^{ncko}* CNCCs missed a developmental window necessary for septation progression.

We have reflected new data in Figs. 3-6, S3 and associated text in “Materials and Methods” on page 19-20, “Results” on pages 7-10 and “Discussion” on pages 12-14.

7. It is unclear why the authors chose to only quantify their phenotypes at the medial level of the outflow tract in figure 7. Also, the pSmad fluorescence intensity should be normalized to the number of YFP+ cells within the ROI and reported as a proportion similar to panel 7N to rule out potential effects due to reduced cell numbers.

Thank you for this suggestion. Since the control distal region does not have OFT cushions due to AP septum formation, we chose to compare the medial regions in both control and *Ankrd11^{ncko}* cushions to have anatomically matched regions. However, we have extended our analysis to distal regions of *Ankrd11^{ncko}* OFT cushions (updated Figs. 4-5, new Figs. 6, S3-4).

We have also quantified pSmad1/5/8 signal by setting a signal intensity threshold and only counting marker+ cells that showed a signal above the threshold (former Fig. 7; now updated Fig. 4). This allowed us to count all YFP+ CNCC cells in the OFT mesenchyme and present the results as a

proportion of total YFP+ cells. Our new results support our original conclusions and show reduced % of pSmad1/5/8+ CNCCs in medial E11.5 *Ankrd11^{ncko}* CNCCs.

We have reflected new data in Figs. 4-5 and associated text in “Materials and Methods” on page 19-20, “Results” on page 7 and “Discussion” on page 12.

8. This reviewer would like to see the raw data that was used to perform statistical analysis to ensure that conditions for using a parametric test on a small sample size were met. The data plotted in the figures suggest unequal variance between samples, which doesn't seem to have been considered in the authors' choice of statistical test.

Thank you for expressing this concern. All IHC data were subjected to normality tests using the Anderson-Darling, D'Agostino & Pearson, and Kolmogorov-Smirnov tests (using Prism) and were found to be normal or have insufficient sample size for these tests and were thus considered normal, except for Fig. S1c, which was then tested using the Mann-Whitney test. Insufficient sample size is typical for biological data as normality tests often require $n > 5$. Our data has typically included $n=3-4$ biological replicates per genotype. Statistical tests and number of biological replicates is reflected this in Materials and Methods on page 21 and in each figure legend. We have also supplied the data in Source files and details on biological replicates, statistical tests and other parameters in the accompanying reporting summary.

Reviewer #3 (Remarks to the Author):

In this study the authors claim that *Ankrd11* is a critical regulator of heart development. *Ankrd11* is a chromatin regulator related to KBG syndrome which clinical manifestation includes, macrodontia, craniofacial findings, short stature, skeletal anomalies, global developmental delay, seizures, and intellectual disability. Heart anomalies were initially described in a minor proportion (Sirmaci, A., et al. *Am. J. Hum. Genet.* 89: 289-294, 2011), but now is found in ~44% of KBG patients (Digilio MC, et al. *Am J Med Genet A.* 2022 Apr;188(4):1149-1159).

• What are the noteworthy results?

The authors showed that ablation of *Ankrd11* in the murine neural crest is related to aberrant heart development and function. This is the main result. In addition, the characterization an abnormal signaling pathways pinpoint the molecular mechanism that may be involved.

• Will the work be of significance to the field and related fields? How does it compare to the established literature? If the work is not original, please provide relevant references.

The work is original. The cardiac defects are more severe than the ones observed in KBG syndrome, but that difference may be related to gene dosage.

We thank the reviewer for providing their constructive feedback and for highlighting the potential of our results. Below we have addressed all raised concerns.

• Does the work support the conclusions and claims, or is additional evidence needed

1. One concern is that the ablation of *Ankrd11* is very detrimental by itself. *Ankrd11* KO animals don't survive. So, maybe the ablation of *Ankrd11* is killing the cardiac neural crest cells. It seems not to be the case since YFP markers are present in the *ncko* embryos, but YFP expression is a reporter for *wnt1* expression, and maybe ectopic expression of *wnt1* is marking other cells.

We thank the reviewer for their feedback. The Wnt1Cre mouse model is the most widely used model for studying cardiac neural crest cells⁴. At E11.5 the YFP distribution in Ankrd11^{n^{ch}et} and Ankrd11^{n^{ck}o} is characteristic of the neural crest lineage (new Fig. 1f-g) and consistent with other publications³⁵, which includes derivatives of the pharyngeal arches and regions alongside the neural tube, which will become dorsal root ganglia of the peripheral nervous system. Therefore, the YFP+ cells in our mouse models are behaving like neural crest cells.

In the OFT, TUNEL staining of YFP+ cells showed no difference in apoptosis between control and Ankrd11^{n^{ck}o} embryos, indicating that apoptosis, which represents major cell death in the developing OFT³⁶, is not a contributing factor in the neural crest dysregulation (Fig. S2a''-f'', former Figs. 3-4).

Furthermore, our results show only minor migration defects of Ankrd11^{n^{ck}o} CNCCs at E10.5 as well as the cells' ability to produce SMA+ smooth muscle cells (updated Fig. S2a-i, Fig. S1j-p). For SMA, we have performed immunofluorescence analysis of cross sections of E18.5 outflow tract vessels and quantified SMA+YFP+ cells out of the total SMA+ cells within the proximal (closest to the heart) and distal regions of the vessels (new Fig. S1j-p). In the proximal region, analysis of the α SMA+ (alpha-smooth muscle actin) smooth muscle layer of the E18.5 Ankrd11^{n^{ck}o} truncal artery did not show a deficiency of CNCC-derived smooth muscle cells compared to the Ankrd11^{n^{ch}et} aortic and pulmonary arteries (new Fig. S1j-p). Although the Ankrd11^{n^{ck}o} distal region showed a decreased proportion of smooth muscle cells that are CNCC derived (% α SMA+YFP+/ α SMA+; new Fig. S1n), this was not due to their decreased number (#YFP+ α SMA+; new Fig. S1o), but rather due to an increase in non-CNCC derived smooth muscle cells (#YFP- α SMA+; new Fig. S1p). Together, these results indicate that YFP+ cells in Ankrd11^{n^{ch}et} and Ankrd11^{n^{ck}o} embryos behave as cardiac neural crest cells, although Ankrd11^{n^{ck}o} CNCCs exhibit delayed condensation and OFT septation.

2. It is known that ablation of neural crest cells resulted in persistent truncus arteriosus, mis-patterning of the great vessels, outflow malalignments, and hypoplasia or aplasia of the pharyngeal glands. If Ankrd11 ablation in the neural crest cells is not killing the cells but is affecting exclusively the cardiovascular development then hypoplasia of thymus, thyroid, and parathyroid glands, should not be present. How are the thymus, thyroid, and parathyroid glands in the ncko animals? If they are normal, this will reinforce that the effect is on cardiac development specifically. I understand that craniofacial abnormalities are also seen in these mice, but those are also seen in KBG syndrome, hence expected.

Thank you for this great suggestion. We were successful in analyzing the sizes of the thymus at E18.5, which did not differ between control and Ankrd11^{n^{ck}o} embryos (page 6, new Fig. S1a-c).

3. Something that maybe problematic is that when you look into the information on B6.Cg-E2f1Tg(Wnt1-Cre)2Sor/J Strain #022501 Common Name: B6 Wnt1-Cre2 <https://www.informatics.jax.org/allele/MGI:5485027?recomRibbon=open>): you find

cardiovascular system

pulmonary trunk hypoplasia (J:298597)

- about 5% of mutants show a hypoplastic pulmonary trunk at E15.5 and E16.5

interrupted aortic arch, type b (J:298597)

- about 30% of mutants exhibit interrupted aortic arch, type b at E15.5 and E16.5

abnormal conotruncal ridge morphology (J:298597)

- the number of neural crest cells in the proximal outflow tract cushions is reduced at E11.5

double outlet right ventricle (J:298597)

- all mutants exhibit the double outlet right ventricle at E15.5 and E16.5
- ventricular septal defect (J:298597)
- all mutants exhibit a ventricular septal defect at E15.5 and E16.5

If <https://www.jax.org/strain/022501> is not the strain used, please clarify.

Thank you for your comment. The phenotypes quoted above were taken from mouse models that used the Wnt1Cre2 driver to ablate other genes. For example, J:298597 refers to a mouse model with this allelic composition: Chd7^{tm2a(EUCOMM)Wtsi}/Chd7^{tm2a(EUCOMM)Wtsi}/E2f1^{Tg(Wnt1-cre)2Sor}/E2f1⁺/Gt(ROSA)^{26Sortm4(ACTB-tdTomato,-EGFP)Luo}/Gt(ROSA)^{26Sor+}. This creates a knockout of the Chd7 allele using the Wnt1Cre2 driver. Upon publication of this manuscript, Ankrd11^{ncko} phenotypes described in this report will be similarly captured in the MGI database.

4. Ankrd11^{nchet} embryos (Ankrd11^{fl/WT}; Wnt1Cre2) showed normal OFT septation, how many embryos were analyzed? There are some haploinsufficiency Ankrd11 related to KBG syndrome presenting heart abnormalities (Digilio MC, et al.. Am J Med Genet A. 2022 Apr;188(4):1149-1159). It is strange that no defect is seen in heterozygous animals.

At E18.5, we have dissected 9 Ankrd11^{WT/WT}, 29 Ankrd11^{nchet}, and 40 Ankrd11^{ncko} embryos, and we found septation defects only in Ankrd11^{ncko} hearts (updated page 6). While we have not observed severe cardiac defects in the Ankrd11^{nchet} mice, it is possible that they may have subtle defects that we were not able to identify using our analysis techniques. Furthermore, multiple cell types contribute to heart development. Ankrd11^{ncko} mice only had Ankrd11 ablation in the neural crest. The effect of Ankrd11 on other heart cell types is still unknown. Therefore, the cardiac phenotype in KBG patients may be compounded by defects in all the heart cell types, leading to a more severe phenotype. This is reflected in Discussion, page 14.

Minor issues:

1. Embryonic hearts at E18.5 were analyzed for anatomical anomalies: all Ankrd11^{ncko} embryos exhibited a persistent truncus arteriosus (PTA). How many?

At E18.5, we have dissected 9 Ankrd11^{WT/WT}, 29 Ankrd11^{nchet}, and 40 Ankrd11^{ncko} embryos, and we found PTA in all Ankrd11^{ncko} hearts.

2. Supplemental Figure 3: not sure that the image for the wt and the Ankrd11^{ncko} embryos are in the same plane, hence comparable.

Thank you for this comment. Due to expansion of the manuscript and since reviewer 2 has also expressed concerns about these images, we have removed these results and associated conclusions from the manuscript to remove confusion.

3. Figure 3 A and D seem to show a different area for the distal part (white squares).

Thank you for the insightful comment. For analysis, all CNCCs in the OFT mesenchyme were counted. However, due to the rarity of TUNEL+ cells, we needed to use different regions of the OFT for magnified images to show these cells.

- Are there any flaws in the data analysis, interpretation and conclusions? Do these prohibit publication or require revision?

If the authors can explain the points raised, I think it is a good work. Main concern is if the phenotype observed (that is very well documented) is related to Ankrd11 ablation.

- Is the methodology sound? Does the work meet the expected standards in your field?

Yes

- Is there enough detail provided in the methods for the work to be reproduced?

Yes

Thank you for this positive feedback. As we replied above, the phenotypes quoted above were taken from mouse models that used the Wnt1Cre2 driver to ablate other genes. For example, J:298597 refers to an mouse model with this allelic composition:

Chd7^{tm2a(EUCOMM)Wtsi}/Chd7^{tm2a(EUCOMM)Wtsi}/E2f1^{Tg(Wnt1-cre)2Sor/E2f1+/Gt(ROSA)26Sortm4(ACTB-tdTomato,-EGFP)Luo/Gt(ROSA)26Sor+}. This creates a knockout of the Chd7 allele using the Wnt1Cre2 driver. Upon publication of this manuscript, Ankrd11^{ncko} phenotypes described in this report will be similarly captured in the MGI database.

References:

- 1 Jiang, X., Rowitch Dh Fau - Soriano, P., Soriano P Fau - McMahon, A. P., McMahon Ap Fau - Sucov, H. M. & Sucov, H. M. Fate of the mammalian cardiac neural crest. *Development* **127** (2000).
- 2 Darrigrand, J. F. *et al.* Dullard-mediated Smad1/5/8 inhibition controls mouse cardiac neural crest cells condensation and outflow tract septation. *Elife* **9**, doi:10.7554/eLife.50325 (2020).
- 3 Liu, X. *et al.* Single-Cell RNA-Seq of the Developing Cardiac Outflow Tract Reveals Convergent Development of the Vascular Smooth Muscle Cells. *Cell Rep* **28**, 1346-1361 e1344, doi:10.1016/j.celrep.2019.06.092 (2019).
- 4 Neeb, Z., Lajiness, J. D., Bolanis, E. & Conway, S. J. Cardiac outflow tract anomalies. *Wiley Interdiscip Rev Dev Biol* **2**, 499-530, doi:10.1002/wdev.98 (2013).
- 5 Zaffran, S., Robrini, N. & Bertrand, N. Retinoids and Cardiac Development. *Journal of Developmental Biology* **2**, 50-71, doi:10.3390/jdb2010050 (2014).
- 6 Keyte, A. L., Alonzo-Johnsen, M. & Hutson, M. R. Evolutionary and developmental origins of the cardiac neural crest: building a divided outflow tract. *Birth Defects Res C Embryo Today* **102**, 309-323, doi:10.1002/bdrc.21076 (2014).
- 7 Stefanovic, S., Etchevers, H. C. & Zaffran, S. Outflow Tract Formation-Embryonic Origins of Conotruncal Congenital Heart Disease. *J Cardiovasc Dev Dis* **8**, doi:10.3390/jcdd8040042 (2021).
- 8 Yan, S., Lu, J. & Jiao, K. Epigenetic Regulation of Cardiac Neural Crest Cells. *Front Cell Dev Biol* **9**, 678954, doi:10.3389/fcell.2021.678954 (2021).
- 9 Ramsdell, A. F. Left-right asymmetry and congenital cardiac defects: getting to the heart of the matter in vertebrate left-right axis determination. *Dev Biol* **288**, 1-20, doi:10.1016/j.ydbio.2005.07.038 (2005).
- 10 Nelms, B. L. & Labosky, P. A. *Transcriptional control of neural crest development.* (Morgan & Claypool Life Sciences, 2010).
- 11 Plein, A., Fantin, A. & Ruhrberg, C. Neural crest cells in cardiovascular development. *Curr Top Dev Biol* **111**, 183-200, doi:10.1016/bs.ctdb.2014.11.006 (2015).

- 12 Nie, X., Ricupero, C. L., Jiao, K., Yang, P. & Mao, J. J. mTOR deletion in neural crest cells disrupts cardiac outflow tract remodeling and causes a spectrum of cardiac defects through the mTORC1 pathway. *Dev Biol* **477**, 241-250, doi:10.1016/j.ydbio.2021.05.011 (2021).
- 13 Hegarty, S. V., O'Keeffe, G. W. & Sullivan, A. M. BMP-Smad 1/5/8 signalling in the development of the nervous system. *Prog Neurobiol* **109**, 28-41, doi:10.1016/j.pneurobio.2013.07.002 (2013).
- 14 Shi, Y. & Massague, J. Mechanisms of TGF-beta signaling from cell membrane to the nucleus. *Cell* **113**, 685-700, doi:10.1016/s0092-8674(03)00432-x (2003).
- 15 Yang, L. *et al.* The mTORC1 effectors S6K1 and 4E-BP play different roles in CNS axon regeneration. *Nat Commun* **5**, 5416, doi:10.1038/ncomms6416 (2014).
- 16 Arenas, D. J. *et al.* Increased mTOR activation in idiopathic multicentric Castleman disease. *Blood* **135**, 1673-1684, doi:10.1182/blood.2019002792 (2020).
- 17 McLennan, R. & Krull, C. E. Ephrin-as cooperate with EphA4 to promote trunk neural crest migration.
- 18 Sanford, L. P. *et al.* TGFbeta2 knockout mice have multiple developmental defects that are non-overlapping with other TGFbeta knockout phenotypes.
- 19 Chen, W. *et al.* Single-cell transcriptomic landscape of cardiac neural crest cell derivatives during development. *EMBO Rep* **22**, e52389, doi:10.15252/embr.202152389 (2021).
- 20 Yamagishi, H. Cardiac Neural Crest. *Cold Spring Harb Perspect Biol* **13**, doi:10.1101/cshperspect.a036715 (2021).
- 21 Theveneau, E. & Mayor, R. Neural crest delamination and migration: from epithelium-to-mesenchyme transition to collective cell migration. *Dev Biol* **366**, 34-54, doi:10.1016/j.ydbio.2011.12.041 (2012).
- 22 Chen, Q. *et al.* Smad7 is required for the development and function of the heart. *J Biol Chem* **284**, 292-300, doi:10.1074/jbc.M807233200 (2009).
- 23 Napoli, J. L. Cellular retinoid binding-proteins, CRBP, CRABP, FABP5: Effects on retinoid metabolism, function and related diseases. *Pharmacol Ther* **173**, 19-33, doi:10.1016/j.pharmthera.2017.01.004 (2017).
- 24 Gandhi, S., Ezin, M. & Bronner, M. E. Reprogramming Axial Level Identity to Rescue Neural-Crest-Related Congenital Heart Defects. *Dev Cell* **53**, 300-315 e304, doi:10.1016/j.devcel.2020.04.005 (2020).
- 25 Saadey, A. A. *et al.* Rebalancing TGFbeta1/BMP signals in exhausted T cells unlocks responsiveness to immune checkpoint blockade therapy. *Nat Immunol* **24**, 280-294, doi:10.1038/s41590-022-01384-y (2023).
- 26 Jaako, P. *et al.* Dietary L-leucine improves the anemia in a mouse model for Diamond-Blackfan anemia. *Blood* **120**, 2225-2228, doi:10.1182/blood-2012-05-431437 (2012).
- 27 Zhang, A. *et al.* Identification of a novel family of ankyrin repeats containing cofactors for p160 nuclear receptor coactivators. *J Biol Chem* **279**, 33799-33805, doi:10.1074/jbc.M403997200 (2004).
- 28 Comai, G. E. *et al.* Local retinoic acid signaling directs emergence of the extraocular muscle functional unit. *PLoS Biol* **18**, e3000902, doi:10.1371/journal.pbio.3000902 (2020).
- 29 Jain, R. *et al.* Cardiac neural crest orchestrates remodeling and functional maturation of mouse semilunar valves. *J Clin Invest* **121**, 422-430, doi:10.1172/JCI44244 (2011).
- 30 Plein, A. *et al.* Neural crest-derived SEMA3C activates endothelial NRP1 for cardiac outflow tract septation. *J Clin Invest* **125**, 2661-2676, doi:10.1172/JCI79668 (2015).
- 31 Toyofuku, T. *et al.* Repulsive and attractive semaphorins cooperate to direct the navigation of cardiac neural crest cells. *Dev Biol* **321**, 251-262, doi:10.1016/j.ydbio.2008.06.028 (2008).
- 32 Kodo, K. *et al.* Regulation of Sema3c and the Interaction between Cardiac Neural Crest and Second Heart Field during Outflow Tract Development. *Sci Rep* **7**, 6771, doi:10.1038/s41598-017-06964-9 (2017).
- 33 Delloye-Bourgeois, C. *et al.* Microenvironment-Driven Shift of Cohesion/Detachment Balance within Tumors Induces a Switch toward Metastasis in Neuroblastoma. *Cancer Cell* **32**, 427-443 e428, doi:10.1016/j.ccell.2017.09.006 (2017).
- 34 Kaartinen, V. *et al.* Cardiac outflow tract defects in mice lacking ALK2 in neural crest cells. *Development* **131**, 3481-3490, doi:10.1242/dev.01214 (2004).

- 35 Lewis, A. E., Vasudevan, H. N., O'Neill, A. K., Soriano, P. & Bush, J. O. The widely used Wnt1-Cre transgene causes developmental phenotypes by ectopic activation of Wnt signaling. *Dev Biol* **379**, 229-234, doi:10.1016/j.ydbio.2013.04.026 (2013).
- 36 Wang, J. *et al.* Defective ALK5 signaling in the neural crest leads to increased postmigratory neural crest cell apoptosis and severe outflow tract defects. *BMC Dev Biol* **6**, 51, doi:10.1186/1471-213X-6-51 (2006).

REVIEWER COMMENTS

Reviewer #1 (Remarks to the Author):

In this revised manuscript, Kibalnyk and colleagues continue to explore the role Ankrd11 in cardiac neural crest cells. As mentioned by Reviewer 2, this is an extension of the group's work published in 2021. I appreciate the effort the authors put into the revised manuscript. The pathways identified as dysregulated in the original manuscript are supported with additional analyses/staining. However, some of their response does not really address the original comments directly.

In figure 1J - I would have expected Ankrd11 to be preserved in non-lineage traced/neural crest-derived cells. However, I do not see any any signal for Ankrd11 in the cells outside the indicated neural crest prongs.... (especially since it is predicted to be broadly expressed by the single cell RNA seq data and external expression datasets). Additional pictures and/or changing the color of the DAPI signal (to generate more contrast between the RNA signal and DAPI) might be helpful. In addition, I appreciate the reduction in pSMAD 1/5/8+ and pS6+ cells in the mutant OFT at E11.5 relative to controls. However, the normalization by E12.5 suggests that perhaps the slight alteration in BMP signaling or mTORC which is normalized may not be relevant to the dramatic septation defect observed.

The MERFISH identifies additional dysregulated pathways and changes in cellular populations. However, my comment originally that the work is mostly descriptive and the MERFISH does not address how Ankrd11 KO causally results in disrupted OFT septation. It is understandable (and perhaps expected) that treating pregnant dams will prove to be difficult and hence my suggestion to consider an explant model to normalize some fraction of relevant gene expression and/or proxy of neural crest behavior (and not necessarily septation). As the authors suggest, it is possible that the control datasets are useful to the community.

Reviewer #2 (Remarks to the Author):

In this revised manuscript, the authors have reframed the conclusions made in the original submission. The revised text describes changes observed in different signaling pathways, and hypothesizes their role in proper OFT septation. The addition of MERFISH data is extremely useful to contextualize the in vivo findings. Overall, I am satisfied by the authors' response to my original comments.

This work will be useful for scientists studying outflow tract development. I commend the authors on their revision experiments.

Reviewer #2 (Remarks on code availability):

While I did not run the code, I scanned through it. The code looks okay to me, although it can benefit from more comments.

Reviewer #3 (Remarks to the Author):

Very interesting and informative work.

Reviewer #1 (Remarks to the Author):

In this revised manuscript, Kibalnyk and colleagues continue to explore the role *Ankrd11* in cardiac neural crest cells. As mentioned by Reviewer 2, this is an extension of the group's work published in 2021. I appreciate the effort the authors put into the revised manuscript. The pathways identified as dysregulated in the original manuscript are supported with additional analyses/staining. However, some of their response does not really address the original comments directly.

In figure 1J - I would have expected *Ankrd11* to be preserved in non-lineage traced/neural crest-derived cells. However, I do not see any any signal for *Ankrd11* in the cells outside the indicated neural crest prongs.... (especially since it is predicted to be broadly expressed by the single cell RNA seq data and external expression datasets). Additional pictures and/or changing the color of the DAPI signal (to generate more contrast between the RNA signal and DAPI) might be helpful.

Thank you for this insightful comment. We apologize for not presenting clear images from various OFT compartments. As requested, we changed the colors and separated channel images for the *Ankrd11* mRNA and DAPI in order to make the *Ankrd11* signal more visible (Fig. 1i-l). We also added additional zoomed in images of the OFT myocardium (Fig. 1i'-l') and neural tube (Fig. 1n-o), which show preserved *Ankrd11* signal in the *Ankrd11^{ncko}* non-neural crest-derived cells. This mirrors successful recombination in CNCCs and absence of recombination in non-neural crest derived tissues like myocardium as assessed by YFP immunostaining (Figs. 3-6). Importantly, our MERFISH results show a reduction in *Sema3c*, a critical regulator of CNCC condensation and OFT septation, only in CNCCs and not OFT myocardium (Fig. S6). Furthermore, we show perturbations in pSmad1/5/8, pS6, pSmad2/3 and CRABP only in E11.5 CNCCs with apparently comparable levels in OFT myocardium (Figs. 4a-h, 5a-h, 6a-h and S4a-h). Finally, human patients with *ANKRD11* loss-of-function variants display heart defects in tissues that are derived from or shaped by the CNCCs, such as dysplastic valves, ventricular septal defects (VSD), aortic coarctation and patent ductus arteriosus ¹. Therefore, our data converge from and are supported by several independent lines of evidence.

We have reflected new data in Fig. 1i-o and associated text in lines 141-145 (“Furthermore, the non-neural crest-derived OFT myocardium and neural tube tissue showed comparable *Ankrd11* signal in control and *Ankrd11^{ncko}* embryos (Fig. 1i'-l', n-o).”)

In addition, I appreciate the reduction in pSMAD 1/5/8+ and pS6+ cells in the mutant OFT at E11.5 relative to controls. However, the normalization by E12.5 suggests that perhaps the slight alteration in BMP signaling or mTORC which is normalized may not be relevant to the dramatic septation defect observed.

Thank you for the comment. Any tissue morphogenesis, including OFT morphogenesis, requires the developmental processes, such as mesenchymal condensation, to occur at precise developmental time windows ²⁻⁶. A delay in gene expression patterns, which control developmental dynamics, has been strongly linked to pathobiological mechanisms of various developmental disorders ²⁻⁵. While OFT condensation delay has not been studied extensively, Darrigand et al. found that *activation of BMP signalling doubled pSmad1/5/8 levels in CNCCs at*

E11.5, leading to *premature CNCC condensation* and asymmetric OFT septation, causing pulmonary stenosis ⁶. Our findings support these results by showing that a *50% reduction in pSmad1/5/8+ cells* (proxy for *BMP signalling*) in Ankrd11^{ncKO} OFT mesenchyme at E11.5 correlated with *delayed CNCC condensation*, since both the pSmad1/5/8 signal and condensation were restored at E12.5. Furthermore, Darrigrand et al. found that the BMP pathway at least in part regulated *Sema3c* expression, a crucial chemoattractant for CNCC condensation and OFT septation, where *overactivation of BMP signaling* in CNCCs caused *increased Sema3C expression*, contributing to an overcondensation phenotype ⁶. Notably, *Sema3c was reduced* in the Ankrd11^{ncKO} OFT in MERFISH and RNA scope results (Figs. S6, 8), correlating with the *impaired condensation* phenotype. This provides evidence that the fine-tuning of BMP signalling is important for OFT condensation, and small changes to the signal level can cause defects in the condensation and consequent septation process. Notably, there are no studies investigating how mTORC1 fine-tuning affects OFT septation. We have highlighted this in the discussion on lines 495-508:

“Neural crest-specific ablation of *Ctdnep1*, a BMP inhibitor, causes an opposing phenotype to the defects observed in the Ankrd11^{ncKO} embryos, specifically premature CNCC condensation and asymmetric AP septum formation, as well as increased pSmad1/5/8 activity ⁶. Furthermore, the BMP pathway was found to at least in part regulate *Sema3c* expression ⁶, a crucial factor for CNCC condensation that was downregulated in the Ankrd11^{ncKO} OFT in MERFISH and RNA scope results. *Sema3C* is a glycoprotein that is secreted within the OFT and pharyngeal arches, and is considered a chemoattractant guidance signal to promote CNCC migration into these structures, their condensation within the cardiac cushions and their migration to create the AP septum ⁶⁻¹⁰. Overactivation of BMP signaling in CNCCs causes increased *Sema3C* expression, contributing to an overcondensation phenotype ⁶. This suggests that the BMP signaling deficiency observed in Ankrd11^{ncKO} CNCCs may at least in part lead to their delayed condensation. Together, this provides evidence that the fine-tuning of BMP signalling is important for OFT condensation, and relatively small changes to the signal level can cause defects in the condensation and consequent septation process.”

The MERFISH identifies additional dysregulated pathways and changes in cellular populations. However, my comment originally that the work is mostly descriptive and the MERFISH does not address how Ankrd11 KO causally results in disrupted OFT septation. It is understandable (and perhaps expected) that treating pregnant dams will prove to be difficult and hence my suggestion to consider an explant model to normalize some fraction of relevant gene expression and/or proxy of neural crest behavior (and not necessarily septation). As the authors suggest, it is possible that the control datasets are useful to the community.

Thank you for this suggestion. We would have also liked to perform explant studies. There are several caveats with this approach that precluded us from pursuing these experiments. First, as we have highlighted in our first response to reviewers, our *in vivo* results do not show differences in CNCC proliferation, apoptosis or migration, which are primary outcome measures in neural crest explants ^{7,11}. Furthermore, we have performed bulk RNA-seq on entire OFT dissected from E11.25 Ankrd11^{Control} and Ankrd11^{ncKO} embryos (data not shown), which did not show statistically significant changes in global gene expression except for three genes whose role in OFT morphogenesis is not known (*Spock1*, *Mctp2*, *Shc4*). This is most probably due to the presence of a large amount of non-neural crest derived cells and/or lack of spatial resolution.

This is supported by our MERFISH results, which show e.g. a reduction in *Sema3c* only in CNCCs and not OFT myocardium (Fig. S6). This further supports the choice of single cell spatial transcriptomics experiments to probe the role of *Ankrd11* in CNCC gene expression (Figs. 7-8, S6).

We thank the reviewer for highlighting the utility of our control datasets for the research community. Our work is also contributing to the field of OFT morphogenesis by chromatin regulators, like *Ankrd11*, and to the understanding of pathobiological mechanisms of KBG syndrome.

References:

- 1 Digilio, M. C. *et al.* Congenital heart defects in molecularly confirmed KBG syndrome patients. *Am J Med Genet A*, doi:10.1002/ajmg.a.62632 (2021).
- 2 Chojnowski, J. L. *et al.* Multiple roles for HOXA3 in regulating thymus and parathyroid differentiation and morphogenesis in mouse. *Development* **141**, 3697-3708, doi:10.1242/dev.110833 (2014).
- 3 Gao, G. *et al.* Isthmin-1 (*Ism1*) modulates renal branching morphogenesis and mesenchyme condensation during early kidney development. *Nat Commun* **14**, 2378, doi:10.1038/s41467-023-37992-x (2023).
- 4 Hinton, R. J. Genes that regulate morphogenesis and growth of the temporomandibular joint: a review. *Dev Dyn* **243**, 864-874, doi:10.1002/dvdy.24130 (2014).
- 5 Kathiriya, I. S. & Srivastava, D. Left-right asymmetry and cardiac looping: implications for cardiac development and congenital heart disease. *Am J Med Genet* **97**, 271-279, doi:10.1002/1096-8628(200024)97:4<271::aid-ajmg1277>3.0.co;2-o (2000).
- 6 Darrigrand, J. F. *et al.* Dullard-mediated Smad1/5/8 inhibition controls mouse cardiac neural crest cells condensation and outflow tract septation. *Elife* **9**, doi:10.7554/eLife.50325 (2020).
- 7 Plein, A. *et al.* Neural crest-derived SEMA3C activates endothelial NRP1 for cardiac outflow tract septation. *J Clin Invest* **125**, 2661-2676, doi:10.1172/JCI79668 (2015).
- 8 Toyofuku, T. *et al.* Repulsive and attractive semaphorins cooperate to direct the navigation of cardiac neural crest cells. *Dev Biol* **321**, 251-262, doi:10.1016/j.ydbio.2008.06.028 (2008).
- 9 Kodo, K. *et al.* Regulation of *Sema3c* and the Interaction between Cardiac Neural Crest and Second Heart Field during Outflow Tract Development. *Sci Rep* **7**, 6771, doi:10.1038/s41598-017-06964-9 (2017).
- 10 Delloye-Bourgeois, C. *et al.* Microenvironment-Driven Shift of Cohesion/Detachment Balance within Tumors Induces a Switch toward Metastasis in Neuroblastoma. *Cancer Cell* **32**, 427-443 e428, doi:10.1016/j.ccell.2017.09.006 (2017).
- 11 He, F. & Soriano, P. A critical role for PDGFRalpha signaling in medial nasal process development. *PLoS Genet* **9**, e1003851, doi:10.1371/journal.pgen.1003851 (2013).